# Efficient High Dimensional Bayesian Optimization with Additivity and Quadrature Fourier Features

**Mojmír Mutný**
Department of Computer Science
ETH Zurich, Switzerland
mojmir.mutny@inf.ethz.ch

**Andreas Krause**
Department of Computer Science
ETH Zurich, Switzerland
krausea@inf.ethz.ch

## Abstract

We develop an efficient and provably no-regret Bayesian optimization (BO) algorithm for optimization of black-box functions in high dimensions. We assume a generalized additive model with possibly overlapping variable groups. When the groups do not overlap, we are able to provide the first provably no-regret *polynomial time* (in the number of evaluations of the acquisition function) algorithm for solving high dimensional BO. To make the optimization efficient and feasible, we introduce a novel deterministic Fourier Features approximation based on numerical integration with detailed analysis for the squared exponential kernel. The error of this approximation decreases *exponentially* with the number of features, and allows for a precise approximation of both posterior mean and variance. In addition, the kernel matrix inversion improves in its complexity from cubic to essentially linear in the number of data points measured in basic arithmetic operations.

## 1 Introduction

Bayesian Optimization (BO) is a versatile method for global optimization of a black-box function using noisy point-wise observations. BO has been employed in selection of chemical compounds [21], online marketing [44], reinforcement learning problems [15, 29], and in search for hyperparameters of machine learning algorithms [25]. BO requires a probabilistic model that reliably models the uncertainty in the unexplored part of the domain of the black-box function. This model is used to define an *acquisiton function* whose maximum determines the next sequential query of black-box function. A popular choice for a probabilistic model is a Gaussian process (GP), a generalization of Gaussian random vector to the space of functions.

BO is very successful when applied to functions of a low dimension. However already problems with $5$ and more dimensions can be challenging for general BO if they need to be optimized efficiently and to a high accuracy. Practical high dimensional BO with GPs usually incorporates an assumption on the covariance structure of a GP, or the black-box function. In this work, we focus on BO with additive GPs [13], and generalized additive GPs [40] with possibly overlapping variable groups allowing cross group interference. Even with the additive models assumption, BO in high dimension remains a daunting task. There are two main problems associated with high dimensional BO with generalized additive GPs, namely, optimization of the acquisition function, and efficient handling of many data points - *large-scale* BO.

To alleviate the two problems, using a generalized additive model assumption, and a popular acquisiton function - Thompson sampling [51], we design efficient no-regret algorithms for solving high dimensional BO problems. Thompson sampling has an acquisition function which leads to a natural block coordinate decomposition in the variable groups when used with additive models without overlapping groups, which reduces the complexity of the acquisition function. In fact, with this assumption, we show that the number of evaluations of the acquisition function are polynomial

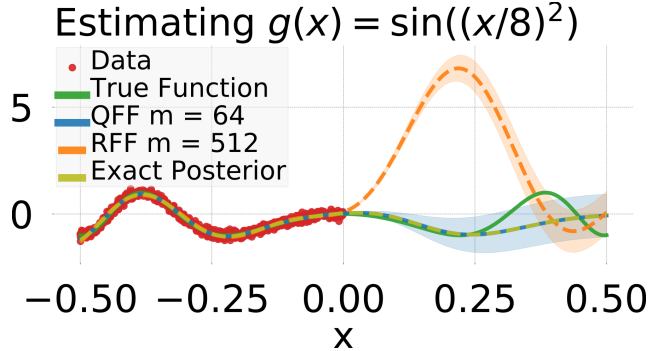

Figure 1: A GP fitted on noisy observations of $g(x)$ with $T = 1024$ data points. One-$\sigma$ confidence bounds are provided in the shaded regions. The parameter $m$ denotes the size of the Fourier basis. RFF cannot produce reliable confidence bounds - variance starvation. On the other hand, QFF do not have this problem, and provide accurate approximation even with a much smaller basis size. The true and approximated confidence intervals intersect exactly in the example above. The example comes from [54].

in the number of points queried during the BO process. The assembly of the acquisition function involves an inversion of kernel matrix and hence scales cubically with the number of data points $T$.

To ensure efficient and scalable optimization up to a high degree of accuracy without the spiraling computational cost, we devise a high fidelity approximation scheme based on Fourier Features and methods from numerical integration. We denote this approximation - *Quadrature Fourier Features* (QFF) in a contrast to Random Fourier Features (RFF) [37]. This scheme approximates a stationary kernel by a linear kernel of a fixed dimensionality in a particularly transformed space. For the ease of exposition we focus our analysis on the squared exponential kernel only, but the methods extend to a broader class of kernels.

The approximation scheme allows us to represent sample paths of a GP in a closed form, and hence optimize them efficiently to high accuracy. Moreover, the uniform approximation error of QFF decreases *exponentially* with the size of the linear basis in contrast to the standard RFF, which decreases with the inverse square root of the basis dimension. However, QFF scale unfavorably with the effective dimensionality of the model making them unsuitable for an arbitrary high dimensional kernel approximation. Their strengths manifest on problems with a low dimension or a low *effective dimension*. In the context of generalized additive models, the effective dimension is the dimension of the largest variable group, which is usually small.

**Previous Works** High dimensional BO with GPs has been considered with assumptions either on the covariance or the black-box function previously. Namely, [57, 12, 8] assume a low dimensional active subspace of the black-box function, and [40, 14, 24, 55] assume (generalized) additive kernels. In [14] authors propose a heuristic to identify the additive structure. However, satisfactory theoretical certificates on cumulative regret or sufficiently practical algorithms with acquisition functions that can be efficiently optimized are lacking. In addition, [9] derives high probability bounds on Thompson sampling with GPs in frequentist setting and [42] in the Bayesian framework framework.

To alleviate the computational cost of kernel methods (and GPs), the machine learning community devised various approximation schemes. Among the plethora of approximations, Nyström Features [32], Random Fourier Features (RFF) [37, 36, 49] or more generally Fourier Features [3, 18, 10], and sparse GP (inducing point methods) [45, 28] stand out.

Inducing point methods is a rich and competitive class of algorithms [52, 60, 19]. Very recently, [35] extends the KISS-GP [60] and shows very accurate posterior sampling with linear complexity (in the number of data points) applied to Bayesian optimization. They utilize Toeplitz structure of covariance matrices and an iterative linear system solver. However, their method is not theoretically analyzed in either posterior moments convergence or cumulative regret in contrast to ours.

The approach most closely related to ours is that of [10] and [53]. Both works use methods from numerical quadrature as well. The former proves exponential convergence for certain types of approximations without providing an explicit construction. The latter considers additive kernels of a

different class. In [2], authors consider an orthogonal direction to achieve the same cost of inversion as QFF from the perspective of linear algebra decomposition.

The kernel approximation in the connection with BO usually focuses on resolving the unfavorable cubic cost of kernel inversion. In this context, approximation schemes for GPs such as RFF and Mondrian features [4] have been used in [56] and [54], respectively. However, [55] demonstrates an adversarial example where RFF cannot reproduce reliably the posterior variance - *variance starvation*. A similar conclusion is found in [35] working with Max Value Entropy Search of [56]. We reproduce this example in Figure 1 and show that QFF (even with smaller basis set) do not suffer from this problem and reproduce the variance with high accuracy. More broadly, sparse GPs and Bayesian Neural Networks as possible approximations of kernels have been considered in literature as heuristics for BO [31, 46, 47].

**Contributions**

- We develop a novel approximation strategy for kernel methods and GPs - *Quadrature Fourier Features* (QFF). This approximation is uniform, and for squared exponential kernel its error provably decreases *exponentially* with the number of features.

- By introducing QFF, the computational cost of the kernel inversion for generalized additive models reduces from $\mathcal{O}(T^3)$ to $\mathcal{O}(T(\log T)^2)$ measured in basic arithmetic operations, where $T$ is the number of data points. This approximation allows the use of BO in large-scale settings and speeds up the sequential calculation significantly.

- We prove that Thompson sampling and GP-UCB [48] algorithms are no-regret when combined with QFF approximation, and for squared exponential kernel the bound is the same as without QFF approximation up to logarithmic factors.

- Using an additive kernel without overlapping groups and Thompson sampling acquisition function, QFF allow us to formulate a practical and provably computationally efficient algorithm for high dimensional BO. This algorithm allows optimization of sample paths for Thompson sampling to an arbitrary precision without the need to iteratively sample from the posterior.

- In the supplementary material we provide a general method to construct QFF for a other stationary kernels.

## 2   Generalized Additive Gaussian Processes and Thompson Sampling

A Gaussian process (GP) is fully characterized by its domain $D \subseteq \mathbb{R}^d$, its prior mean (assumed to be zero here), and its kernel function $k : D \times D \to \mathbb{R}$. It is a stochastic process whose all finite marginals are Gaussians, in particular, $f(x) \sim \mathcal{N}(\mu(x), \sigma(x)^2)$, where $\mu(x)$ is the mean and $\sigma(x)^2$ is the variance. The covariance structure of the stochastic process is governed by the kernel function $k(x, y)$.

**Generalized Additive GPs**   Generalized additive models [40] are a generalization of additive models [16] that decompose a function to a sum of functions $g_j$ defined over low-dimensional components. Namely,

$$g(x) = \sum_{j=1}^{G} g^{(j)}(x^{(j)}), \tag{1}$$

where each $x^{(j)}$ belongs to a low dimensional subspace $\mathcal{X}^{(j)} \subseteq D$. With $G$, we always denote the number of these components. Additive models, in contrast to generalized additive models, imply that $\mathcal{X}^{(j)} \cap \mathcal{X}^{(k)} = \emptyset$ if $k \neq j$. In our work, we start with a generalized additive models and specialize to additive models when needed.

The concept of additive models can be extended to Gaussian processes, where the stochastic process $f$ is a sum of stochastic processes $f = \sum_{j=1}^{G} f_j$ where each has low dimensional indexing (dimensions) [13, 39]. With the additive assumption, the kernel and the mean function of an generalized additive GP decomposes in the same fashion as the components $f_j$. Namely, $k(x, y) = \sum_{j=1}^{G} k^{(j)}(x^{(j)}, y^{(j)})$ and $\mu(x) = \sum_{j=1}^{G} \mu^{(j)}(x^{(j)})$. This simplifies the GP, and we

define the *effective dimensionality* of the model as the largest dimension among all additive groups, $\bar{d} = \max_{j \in [G]} \dim(\mathcal{X}^{(j)})$. Next, we explain how these methods can be exploited with BO.

**BO with Posterior sampling**    BO sequentially generates points where the black-box function $g(x)$ should be queried. These points are maximizers of an acquisition function [7]. A popular class of stochastic acquisition functions without a generally tractable closed-form expression is Thompson sampling [51]. In Thompson sampling, a sample from the posterior GP is chosen as the acquisition function at each step.

Using the generalized additive assumption, the sample from a GP ($f \sim$ GP) decomposes as $f(x) = \sum_{j=1}^{G} f_j(x^{(j)})$. With the additive model assumption (no overlapping groups), the individual functions depend on their specific variable groups only, i.e., $\mathcal{X}^{(j)}$. Consequently, $f_j(x^{(j)})$ can be optimized independently on a lower dimensional subspace. Due to this decomposition, Thompson sampling is a natural candidate for BO with additive models. However, the use of Thompson sampling in practice is limited by the computational problems associated with sampling from the posterior.

**The maximum of a sample path**    Principally, a sample path from a GP can be optimized using three methods. The first, *direct method*, samples a path over the whole finite domain at once and finds the maximum. The standard way to sample on a discrete domain $D$ is to perform a Cholesky decomposition of the covariance matrix, which costs $\mathcal{O}(|D|^3)$ basic arithmetic operations. Having a finely discretized domain $D$, this cost might be prohibitive, especially considering that $|D|$ grows exponentially with the dimension. With the additive assumption (non-overlapping), the variable groups are independent, thus one could sequentially sample the GP only on $\mathcal{X}^{(j)} \subseteq D$, and condition on these observations while iterating over groups. However, the method is sequential and requires re-computation of the posterior after each variable groups has been sampled. We refer to this method as *canonical* Thompson sampling with additive models in our benchmarks.

The second option is to sample *iteratively*. Here, we sample a value of the stochastic process at a point and condition on it to sample the next one. With this approach we can optimize the acquisition function over a continuous domain. However, at every single iteration, a new posterior has to be recomputed, which can again be prohibitively slow. The third approach is to use a finite basis approximation. Fourier Features provide such an approximation, and their use is subject of Section 4, where we introduce a closed form expression for the sample paths and their derivatives.

## 3   Fourier Features for Bayesian Optimization

**Bayesian Optimization and Uniform Approximation**    BO requires that the probabilistic model is a reliable proxy for uncertainty. In order to have a method which can truly and faithfully explore the domain of the function, we need that the approximation to the uncertainty model is valid on the whole optimization domain. Consequently, one requires a uniform approximation guarantee. Such guarantees cannot be easily obtained by methods based on past observations such as Nyström features [58], or other adaptive weighting methods unless the obtained data cover the whole domain. As the purpose of BO is to efficiently probe the black-box function, these methods are not compatible with the goal of BO.

One of the popular methods that uniformly approximate the kernel with theoretical guarantees is the Fourier Features method. This approach is applicable to any continuous stationary kernel. According to Bochner's theorem [41], any such kernel can be expressed as a Fourier integral of a dual function $p(\omega)$ in the frequency space. Approximating this integral in a suitable manner can provide a uniform approximation.

**Definition 1** (Uniform Approximation). *Let $k : D \times D \to \mathbb{R}$ be a stationary kernel taking values from $D \subset \mathbb{R}^d$, then the inner product $\Phi(x)^\top \Phi(y)$ in $\mathbb{R}^m$, $\epsilon$-uniformly approximates $k$ if and only if,*

$$\sup_{x,y \in D} |k(x,y) - \Phi(x)^\top \Phi(y)| \leq \epsilon. \tag{2}$$

In Definition 1, generally, $\epsilon$ has a functional dependence on $m$, the size of approximating basis. For example, $\epsilon(m) = \mathcal{O}\left(m^{-1/2}\right)$ for Random Fourier Features. Our analysis reveals that the error of

the uniform approximation translates to the approximation guarantee on posterior mean, posterior variance, and on the cumulative regret for common BO algorithms.

## 3.1 General Fourier Features

Bochner's theorem states the existence of an integral representation for the kernel function, which can be subsequently approximated via a finite sum.

$$k(x-y) \overset{\text{Bochner's thm.}}{=} \int_\Omega p(\omega) \begin{pmatrix} \cos(\omega^\top x) \\ \sin(\omega^\top x) \end{pmatrix}^\top \begin{pmatrix} \cos(\omega^\top y) \\ \sin(\omega^\top y) \end{pmatrix} d\omega \overset{\text{Fourier F.}}{\approx} \Phi(x)^\top \Phi(y) \qquad (3)$$

The finite sum approximation is performed such that each term in the sum is a product of two analytically identical terms, each depending on either $x$ or $y$. This finite sum, in effect, defines a linear kernel in a new space via the mapping $\Phi$. One of the approximations satisfying these requirements is Monte Carlo sampling according to the distribution $p(\omega)$. This is the approximation used for the celebrated Random Fourier Features (RFF) [37, 36, 3].

Linear kernels are desirable as they can be dealt with efficiently. They have a fixed dimensionality, and the inversion of the kernel matrix scales with the dimension of the space rather than the number of data points, as is demonstrated in the next paragraph.

**The Posterior with Fourier Features** We denote the dimension of the Fourier Feature mapping in (3) with $m$. Then the covariance in this approximating linear space is defined by the following quantities. Let $\Phi(\mathbf{X}_t) = (\Phi(x_1), \dots \Phi(x_t))^\top \in \mathbb{R}^{m \times t}$ , then

$$\mathbf{\Sigma}_t = (\Phi(\mathbf{X}_t)^\top \Phi(\mathbf{X}_t) + \rho^2 \mathbf{I}) \quad \text{and} \quad \nu_t = (\mathbf{\Sigma}_t)^{-1} \Phi(\mathbf{X}_t)^\top y \qquad (4)$$

where $\rho$ denotes the additive Gaussian noise incurred to the observations $y$ of the true black-box function $g(x)$. The approximated posterior mean then becomes $\tilde{\mu}_t(x) = \Phi(x)^\top \nu_t$ and the posterior variance $\tilde{\sigma}_t(x)^2 = \rho^2 \Phi(x)^\top \mathbf{\Sigma}_t^{-1} \Phi(x)$, when $\|\Phi(x)\|_2 = 1$ (which is true for RFF and QFF).

## 3.2 Quadrature Fourier Features (QFF)

The literature on Fourier Features concentrates mostly on Random Fourier Features that use Monte Carlo approximation of the integral. In this work, we take the perspective of numerical integration to approximate the integral, and review the basics of numerical quadrature here. Subsequently, we use Hermite-Gauss quadrature (a standard technique in numerical integration) to provide a uniform approximation over $D$ for the squared exponential kernel - *Quadrature Fourier Features* (QFF) with exponentially decreasing error on the uniform approximation.

**Numerical Quadrature** A quadrature scheme for an integral on a real interval is defined by two sets of points - weights and nodes. Nodes are points in the domain of the integrand at which the function is evaluated ($\{\omega_j\}_{j=1}^m$). Weights ($\{v_j\}_{j=1}^m$) are the scaling parameters that scale the evaluations at the nodes. In addition, the integral is usually formulated with a weight function $w(x)$ that absorbs badly behaved properties of the integrand. For further details we refer the reader to the standard literature on numerical analysis [22]. An extension to multiple dimensions can be done by so called Cartesian product grids (Def 2). Cartesian product grids grow exponentially with the number of dimensions, however for small dimensions they are very effective.

**Definition 2** (Cartesian product grid). *Let $D = [a, b]^d$, and $B$ be the set of nodes of a quadrature scheme for $[a, b]$. Then the Cartesian product grid $B^d = B \times B \cdots \times B$, where $\times$ denotes the Cartesian product.*

**Assumption 1** (Decomposability). *Let $k$ be a stationary kernel defined on $\mathbb{R}^d$, s.t. $k(x, y) \leq 1$ for all $x, y \in \mathbb{R}^d$ with Fourier transform that decomposes product-wise $p(\omega) = \prod_{j=1}^d p_j(\omega_j)$.*

**QFF** In order to define QFF we need Assumption 1. This assumption is natural, and is satisfied for common kernels such as the squared exponential (even ARD after the change of variables) or the modified Matérn kernel. Further details can be found in the supplementary material.

**Definition 3** (QFF). *Under Assumption 1 let $m = (2\bar{m})^d$, where $\bar{m} \in \mathbb{N}$. Suppose that $x, y \in [0,1]^d$. Let $p(\omega) = \exp\left(-\sum_{j=1}^{d} \frac{\omega_j^2 \gamma_j^2}{2}\right)$ be the Fourier transform of the kernel $k$. Then, we define the mapping,*

$$\Phi(x)_j = \begin{cases} \sqrt{\prod_{i=1}^{d} \frac{1}{\gamma_j} v(\omega_{j,i})} \cos((\omega_j)^\top x) & \text{if } j \leq m \\ \sqrt{\prod_{i=1}^{d} \frac{1}{\gamma_j} v(\omega_{j-m,i})} \sin((\omega_{j-m})^\top x) & \text{if } 2m > j > m \end{cases}, \tag{5}$$

*where $v(\omega_{j,i}) = \frac{\sqrt{2}}{\gamma_i} \frac{2^{m-1} m! \sqrt{\pi}}{m^2 H_{m-1}(\omega_{j,i})^2}$ and $H_i$ is the ith Hermite polynomial. The set $\{\omega_j\}_{j=1}^{m}$ is formed by the Cartesian product of $\{\bar{\omega}_i\}_{i=1}^{\bar{m}}$, where each element is in $\mathbb{R}$ and, is defined to be the zero of the i-th Hermite polynomial. See Gauss-Hermite quadrature in [22].*

The general scaling of $m$ with dimension $d$ is exponential due to the use of Cartesian grids, however our application area - BO usually involves either small dimensional problems up to 5 dimensions, or high dimensional BO with low *effective dimensions* - generalized additive models - where these methods are very effective.

**Additive kernels**    When using generalized additive kernels $k(x,y) = \sum_{j=1}^{G} k(x^{(j)}, y^{(j)})$, we can use QFF to approximate each single component independently with mapping $\Phi^{(j)}(x^{(j)})^\top \Phi^{(j)}(y^{(j)})$, with $m_j$ features, and stack them together to one vector $\Xi$. In this way, the number of features needed scales exponentially only with the effective dimensions $\bar{d}$, which is usually small even if $d$ is large.

**Approximation Error**    We provide an upper bound on the error of uniform approximation guarantee that decreases exponentially with $m$.

**Theorem 1** (QFF error). *Let $\Phi(x) \in \mathbb{R}^m$ with $m = (2\bar{m})^d$ be as in Definition 3, with inputs in $D = [0,1]^d$ and $\gamma = \min_i \gamma_i$,*

$$\sup_{x,y \in D} |k(x,y) - \Phi(x)^\top \Phi(y)| \leq d2^{(d-1)} \frac{\bar{m}! \sqrt{\pi}}{2^{\bar{m}} (2\bar{m})!} \left(\frac{\sqrt{2}}{\gamma}\right)^{2\bar{m}} \leq d2^{(d-1)} \sqrt{\frac{\pi}{2}} \frac{1}{\bar{m}^{\bar{m}}} \left(\frac{e}{4\gamma^2}\right)^{\bar{m}}. \tag{6}$$

Theorem 1 implies that if $\gamma$ is very small, the decrease might be exponential only for $m > \gamma^{-2}$. This is confirmed by our numerical experiment in Figure 2c, and the break point $m^* = \gamma^{-2}$ at the intersection of the two purple lines predicts the start of the exponential decrease. The error on the posterior mean with this approximation can be seen in Figures 2a and 2b. The exponential decrease of posterior mean with QFF follows from a Theorem 5 in supplementary material.

Furthermore, for general additive kernels, the error bound in Theorem 1 depends only on the effective dimension $\bar{d}$, although the dimension $d$ might be much larger. The fact that additive assumption improves the error convergence can be seen in Figure 2d, where different models with different effective dimensionalities are presented. However, for all models, the dimensionality $d = 3$ stays constant. The approximation has desirable properties even if the variables overlap as the *Circular* example shows. The only requirement on efficiency is the low *effective dimensionality*.

## 4   Efficient Algorithm for High Dimensional Bayesian Optimization

**Thompson Sampling**    Using Thompson sampling (TS) with Fourier Features approximation, we are able to devise an analytic form for the acquisition function. Namely, a sample path from the approximating GP amounts to sampling a fixed dimensional vector $\theta_t \sim (\nu_t, (\Sigma_t)^{-1})$, where quantities come from (4). The rule for Thompson sampling with a generalized additive kernel becomes

$$x_{t+1} = \arg\max_{x \in D} \Xi(x)^\top \theta_t = \arg\max_{x \in D} \sum_{j=1}^{G} \Phi^{(j)}(x^{(j)})^\top \theta_t^{(j)}. \tag{7}$$

Since $\theta_t$ has a fixed dimension $m$, the cost to compute the posterior and the sample path is constant $\mathcal{O}(m^3)$, in contrast to $\mathcal{O}(t^3)$ and $\mathcal{O}(|D|^3)$ for the canonical TS. In addition, this formulations allows the use of first-order optimizers to optimize the acquisition function effectively. The

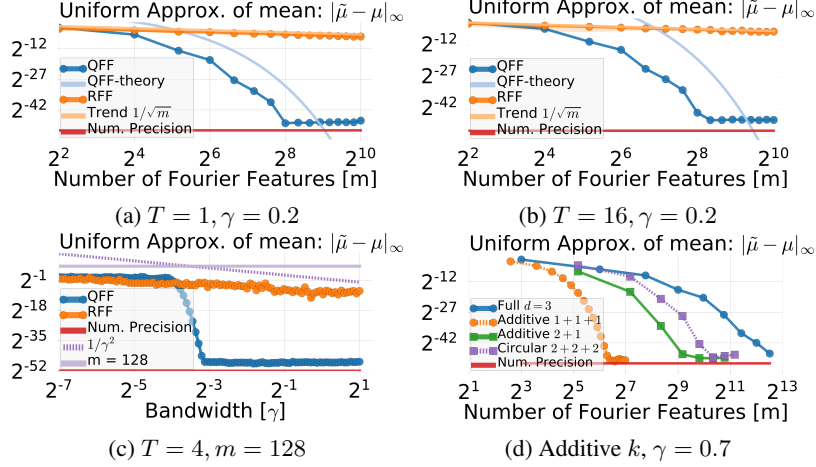

(a) $T = 1, \gamma = 0.2$    (b) $T = 16, \gamma = 0.2$

(c) $T = 4, m = 128$    (d) Additive $k, \gamma = 0.7$

Figure 2: The plots show the error on uniform approximation of the posterior mean estimate. The black-box function $g$ is a sample from GP with squared exponential kernel. For 2a and 2b $d = 2$, for 2c $d = 1$, and 2d $d = 3$ (but some are additive). The tilde denotes the approximated quantities with Fourier Features. The parameter $T$ represents the number of data points. In 2d *Circular* corresponds to overlapping groups $\{(x_1, x_2), (x_2, x_3), (x_3, x_1)\}$ and *Additive* to two non-overlapping groups $\{(x_1, x_2), (x_3)\}$.

acquisition function for each variable group $j$ is Lipschitz continuous with the constant $L_t^{(j)} = \left\| \theta_t^{(j)} \right\| \sqrt{\sum_{i=1}^{m_j/2} 2 v_i^2 \left\| \omega_i \right\|^2}$, thus we can run a global optimization algorithm to optimize the acquisition function presented in (7) provably. Furthermore, optimization to a finer accuracy does not require re-sampling or iterative posterior updates, and can be done adaptively with first-order optimizers or global optimization methods such as DIRECT [23] due to the availability of the analytic expression once $\theta_t$ has been sampled.

With the assumption on additivity (without overlapping groups), the optimization problem in (7) decomposes over variable groups. Hence, one can perform block coordinate optimization independently. For global optimization algorithms, we are able to provide a polynomial bound on the number of evaluations of the acquisition function for a fixed horizon $T$ of queries to the black-box function.

**Theorem 2** (Polynomial Algorithm). *Let* $\delta \in (0, e^{-1})$, $T \in \mathbb{N}$ *be a fixed horizon for BO, $k$ be an additive squared exponential kernel with $G$ groups and $\bar{d}$ - maximal dimension among the additive components. Moreover, let $\Phi^{(j)}(\cdot) \in \mathbb{R}^{m_j}$ be the approximation of the $j$th additive component as in Definition 3 with $m_j \geq 2 \log_\eta (T^3)^{d_j}$ and $m_j \geq \frac{1}{\gamma_j^2}$, where $\eta = 16/e$. Then a Lipschitz global optimization algorithm [34] requires at most*

$$\mathcal{O}\left( \frac{G \log(T/\delta)^{\bar{d}/2}}{\alpha^{\bar{d}}} \left( T^{3/2} (\log T)^{\bar{d}} + T^2 (\log T)^2 \right)^{\bar{d}} \right) \tag{8}$$

*evaluations of the acquisition function (7) to reach accuracy $\alpha$ for each optimization subproblem with probability $1 - \delta$.*

In addition, when the kernel is fully additive ($\bar{d} = 1$) the number of evaluations is at most $\mathcal{O}\left( \frac{d \sqrt{\log(T/\delta)}}{\alpha} \left( T^2 (\log T)^2 \right) \right)$. In practice thanks to the analytic formulation, one can perform gradient ascent to optimize the function with effectively constant work per iteration.

The polynomial algorithm is stated in full in Algorithm 1 with arbitrary Lipschitz global optimization oracle. Note that by design, first the correlated $\theta_t$ is sampled and only then is the acquisition function decomposed and optimized in parallel. This ensures that we include the cross correlation of additive groups, and yet decompose the acquisition function, which has been an open problem of Add-GP-UCB type algorithms [40].

---

**Algorithm 1** Thompson sampling with Fourier Features and additive models

---

**Require:** Fourier Feature mapping $\Phi^{(j)}(x) \in \mathbb{R}^{m_j}$ for each $j \in [G]$, $\alpha_t$ accuracy
**Ensure:** Domain $D = [0,1]^d$, $m_j > \frac{1}{\gamma_j^2}$

   **for** $t = 1, \ldots, T$ **do**
      Update $\nu_t$ and $\boldsymbol{\Sigma}_t$ according to (4).               ▷ Calculate posterior
      Sample $\theta_t \sim \mathcal{N}(\nu_t, (\boldsymbol{\Sigma}_t)^{-1})$         ▷ Sampling via Cholesky decomp.
      **for** $j = 1, \ldots, G$ **do**            ▷ Iterate over the variable groups
         Find $x_t^{(j)} = \arg\max_{x \in D} (\theta_t^{(j)})^\top \Phi^{(j)}(x^{(j)})$       ▷ global optimization
      **end for**
      Query the function, i.e. $y_t = g(x_t) + \epsilon_t$.
   **end for**

---

**Other Acquisition Functions** Apart from Thompson sampling one can apply QFF to significantly improve sampling based acquisition functions such as Entropy Search (ES) [17], Predictive Entropy Search (PES) [20] and Max-Value Entropy Search (MES) [55]. We focus on TS exclusively as the evaluations of the acquisition function is computationally more efficient. In the former methods, one needs to create a statistics describing the maximizer or maximum of the Gaussian process via sampling.

## 5 Regret Bounds

A theoretical measure for BO algorithms is the cumulative regret $R_T = \sum_{t=1}^{T} g(x^*) - g(x_t)$, which represents the cost associated by not knowing the optimum of the black-box function $g(x^*)$ a priori. Usually one desires algorithms that are no-regret, meaning that $\frac{R_T}{T} \to 0$ as $T \to \infty$. In this work, we focus on algorithms with a fixed horizon $T$, and where observations of $g(x)$ are corrupted with Gaussian noise $\epsilon \sim \mathcal{N}(0, \rho^2)$. We provide bounds on the cumulative regret for Thompson sampling and GP-UCB (in appendix).

In the supplementary material, we provide a general regret analysis assuming an arbitrary $\epsilon(m)$-uniformly approximating kernel. This allows us to identify conditions on the dependence of $\epsilon(m)$ such that an algorithm can be no-regret. RFF in contrast to QFF do not achieve sublinear cumulative regret with our analysis. For the exponentially decreasing error of QFF, we can prove that asymptotically our bound on the cumulative regret coincides (up to logarithmic factors) with the bound on *canonical* Thompson sampling in the following theorem (similar result holds for UCB-GP). Our proof technique relies on the ideas introduced in [9].

**Theorem 3.** *Let $\delta \in (0, e^{-1})$, $k$ be additive squared exponential kernel with $G$ components, and the black box function is bounded in RKHS norm. Then running Thompson sampling with the approximated kernel using* QFF *from Definition (3) s.t. $m_j \geq 2(\log_\eta(T^3))^{d_j}$, and $m_j > \gamma_j^{-2}$ for each $j \in [G]$, where each acquisition function is optimized to the accuracy $\alpha_t = \frac{1}{\sqrt{t}}$ suffers a cumulative regret bounded by,*

$$R_T \leq \mathcal{O}\left( G(\log T)^{\bar{d}+1} \sqrt{T} \log\left(\frac{T}{\delta}\right)^{3/2} \right) \tag{9}$$

*with probability $1 - \delta$, where $\bar{d}$ is the dimension of largest additive component.*

Theorem 3 implies that the size of Fourier basis for QFF needs to scale as $m = \mathcal{O}(\log T^3)^{d_j}$ to have a no-regret algorithm. Hence the kernel inversion for $d = 1$ in (4) needs only $\mathcal{O}(T(\log T^3)^2)$ basic arithmetic operations, which can significantly speed up the posterior calculation for low dimensional or low effective dimensional kernels, since for these we have $m = \sum_{j=1}^{G} m_j$.

## 6 Experimental Evaluation

**Benchmark functions** We present cumulative regret plots for standard benchmarks with the squared exponential kernel (Figure 3). We test Thompson sampling with QFF for a fixed horizon with high-dimensional functions used previously in [14]. Details of the experiments are in the

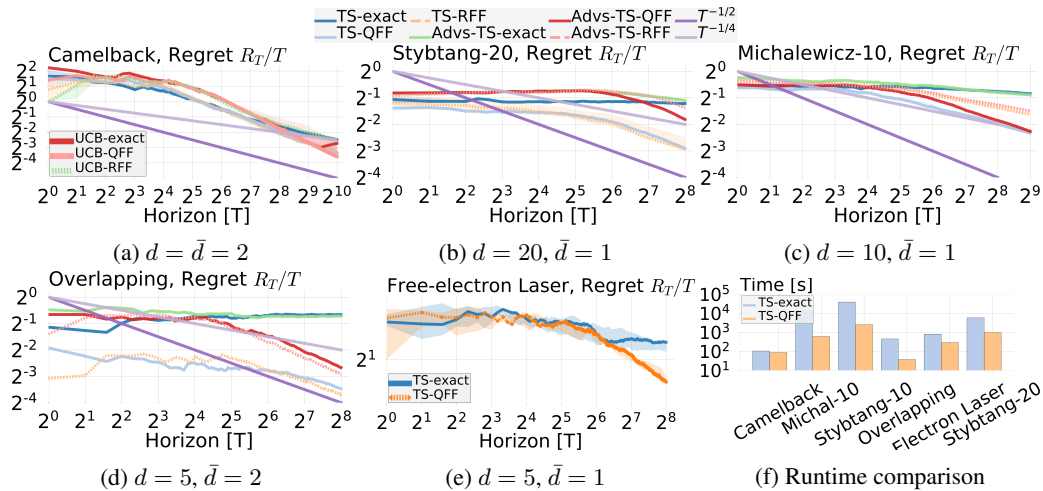

Figure 3: In these graphs we compare exact Thompson sampling (TS-exact), RFF approximation (TS-RFF) and QFF approximation (TS-QFF). We plot the cumulative regret divided by the horizon (iteration) $T$, similar to statements in our theorems. The prefix *Advs* suggests that we started with a set of observations larger than the Fourier basis located in a selected (negative) part the domain in the spirit of the example in Figure 1. For every experiment the full dimension $d$ and the dimension of the largest additive component $\bar{d}$ is specified. The functional forms can be found in the supplementary material.

supplementary material. We compare QFF, RFF and the exact GP. In Figure 3f, we show that for each experiment the speed of computation improves significantly even though for high dimensional experiments the grid for Lipschitz optimization was twice as fine as for the exact method. In some instances the QFF performs better than the BO with exact GP. We hypothesize that in these cases QFF serves as a regularizer and simplifies the BO problem; or in the case of high dimensional functions, we were able to optimize the function with a finer grid than the non-exact method. In addition, RFF perform well on experiments without adversarial initialization, which suggests that on average this approximation can seem to work, but there are adversarial cases like in Figure 3c, where RFF fail.

**Tuning free electron laser** In Figure 3e we present an experiment on real-world objective. This experiment presents preliminary results on automatic tuning of hyperparameters for a large free electron laser SwissFEL located at PSI in Switzerland. We run our algorithm on a simulator that was fit with the data collected from the machine. In the fitting, we used the additive assumption. The simulator reliably models the considerable noise level in the measurements. This experiment is an unusual example of BO as measurements can be obtained very quickly at frequencies up to 1 Hz. However, the results are representative for only a couple of hours due to drift in the system. Therefore, the desire for a method which has an acquisition function that can be efficiently optimized in high dimensions is paramount. The cost of the optimization with our method is fixed and does not vary with the number of data points. Due to very noisy evaluations, the number of queries needed to achieve the optimum is considerable. Our method is the only method which fulfills these criteria, and has provable guarantees. We show that the runtime of our algorithm is an order of magnitude lower than the canonical algorithm, and reaches better solutions as we can afford to optimize the acquisition function to higher accuracy.

# 7 Conclusion

We presented an algorithm for high dimensional BO with generalized additive kernels based on Thompson sampling. We show that the algorithm is no-regret and needs only a polynomial number of evaluations of the acquisition function with a fixed horizon. In addition, we introduced a novel deterministic Fourier Features based approximation of a squared exponential kernel for this algorithm. This approximation is well suited for generalized additive models with a low effective dimension. The approximation error decreases exponentially with the size of the basis for the squared exponential kernel.

## Acknowlegements

This research was supported by SNSF grant 407540_167212 through the NRP 75 Big Data program. The authors would like to thank Johannes Kirschner for valuable discussions. In addition, we thank SwissFEL team for provision of the preliminary data from the free electron laser. In particular we thank Nicole Hiller, Franziska Frei and Rasmus Ischebeck of Paul Scherrer Institute, Switzerland.

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
