[Supplementary Material]

# Contents

# A   Supplementary results - Quadrature Fourier Features

[1]

## A.1   Basic definitions

This subsection contains basic definitions and assumptions needed for further results and proofs.

**Definitions**

**Definition 4** (Spectral characteristic function). *Let $k$ be a stationary kernel and $g \in \mathcal{H}_k$ (associated RKHS) s.t. $g(x) = \sum_{j \in I} \alpha_j k(x - x_j)$, then spectral characteristic function of $g$ is*

$$\alpha(\omega) \stackrel{def}{=} \sum_{j \in I} \alpha_j \exp(i \omega^\top x_j). \tag{10}$$

**Definition 5** (Approximating Space). *Let $\mathcal{F}_m$ be an approximating space defined via the basis functions in the vector $\Phi$, s.t.*

$$\mathcal{F}_m(\Phi) \stackrel{def}{=} \{f : \mathbb{R}^d \to \mathbb{R} \text{ s.t. } f(x) = \psi^\top \Phi(x) | \psi \in \mathbb{R}^m\}, \tag{11}$$

*where the vector $\Phi(x) \in \mathbb{R}^m$, and $\Phi(x)_j = \phi(\omega_j, x)$ for some $\omega_j \in [m]$.*

**Definition 6** (Total Variation). *Let $f : D \to \mathbb{R}$, where $D \subset \mathbb{R}$ be a differentiable function then total variation of $f$ is defined to be,*

$$\mathrm{TV}(f) = \int_D |f'(x)| dx. \tag{12}$$

**Definition 7** (Information Gain). *Let $k$ be a kernel defining the posterior variance $\sigma_t(x)^2$, and $\{x_t\}_{t=1}^T$ the sequence of points chosen. We define the information gain of these points as, $I(\{x_t\}_{t=1}^T, \rho) \overset{def}{=} \frac{1}{2} \sum_{t=1}^T \log(1 + \rho^{-2}\sigma_{t-1}^2(x_t))$, and the maximum information gain as, $\gamma_T(k) \overset{def}{=} \max_{\{x_t\}_{t=1}^T} I(\{y_t\}_{t=1}^T, \rho)$.*

### Assumptions

**Assumption 2** (Uniform Approximation). *Let $\Phi(x)^\top \Phi(y)$ $\epsilon$-uniformly approximate the stationary kernel $k$ on $D = [0,1]^d$ as in (Def 1).*

**Assumption 3** (Bound). *Let $g \in \mathcal{H}$, an RKHS defined via stationary kernel $k$, such that $\|g\|_{\mathcal{H}} \le B$.*

**Assumption 4** (Bounded spectral characteristic). *Let $g$ be a function from $\mathcal{H}_k$, RKHS induced via kernel $k$, s.t. its characteristic spectral function $\alpha(\omega)$ is bounded by $B$ i.e. $|\alpha(\omega)| \le B$.*

**Lemma 1.** *Assumption 4 implies $\|g\|_{\mathcal{H}_k} \le B$ and that further implies $|g(x)| \le B$.*

### A.2 Extensions of QFF - Kernel specific quadrature

First, we note that the there exist a kernel dependent way to approximate the integral that we refer to as *kernel specific*. Namely, we use the fact that the Fourier transform $p(\omega)$ can be used as weighting function in numerical integration. However, not all functions can be valid weighting function on an unbounded domain.

They have to satisfy the following criteria [50]:

1. $p(\omega)$ is measurable on a unbounded interval

2. All moment of $p(\omega)$ are finite

3. for polynomials $s(\omega)$ that are positive $\int p(\omega)s(\omega)d\omega = 0$ implies $s(\omega) = 0$.

These conditions are clearly satisfied for squared exponential kernel, and we provided a specialized quadrature based Fourier Features in Def. 3. There are possibly other kernels satisfying these conditions that can be created as combinations of the above kernels with other kernel functions. A elegant framework that can ensure these properties to hold is to design the kernels in the frequency domain as in [59].

However these conditions cannot be satisfied for Laplace kernel, for example. Its Fourier transform has unbounded moments. For kernels where *kernel specific* Fourier Features cannot be constructed, we present a universal method which requires transformation to a bounded domain in (8), however the exponential error decrease cannot be ensured in general anymore.

**Integration Interval** When specialized quadrature methods defined over unbounded domain cannot be applied, we apply a transformation techniques as outlined in [6] to make the integration domain bounded. There are several way to proceed; we focus on the transformation from [5], $\omega = \cot(\phi)$ due to its elegance and good convergence properties when the function is sufficiently regular. Applying the same transformation coordinate-wise on each component of $\phi$, we arrive at the following integral,

$$k(x - y) = \int_{[0,\pi]^d} p(\cot(\phi))\phi(\cot(\phi), x)\phi(\cot(\phi), y) \prod_{i=1} \frac{1}{\sin(\phi_i)^2} d\phi. \tag{13}$$

**General QFF**

**Definition 8** (General QFF ). *Let Assumption 1 be satisfied, and $m = (2\bar{m})^d$, where $\bar{m} \in \mathbb{N}$, then for $x \in [0,1]^d$, we define a mapping*

$$\Phi(x)_j = \begin{cases} \sqrt{p(\cot(\phi_j))\left(\prod_{i=1}^d \frac{v_j(\phi_{j,i})}{\sin^2(\phi_{j,i})}\right)} \cos(\cot(\phi_j)^\top x) & \text{if } j \leq m \\ \sqrt{p(\cot(\phi_{j-m}))\left(\prod_{i=1}^d \frac{v_j(\phi_{j-m,i})}{\sin^2(\phi_{j-m,i})}\right)} \sin(\cot(\phi_{j-m})^\top x) & \text{if } 2m > j > m, \end{cases} \tag{14}$$

*where $v(\phi_{j,i}) = \frac{2}{(1-\phi_{j,i})^2 L'_{m-1}(\phi_{j,i})^2}$ and $L'_i$ is the derivative of ith Legendre polynomial. The set $\{\phi_j\}_{j=1}^m$ is formed by the Cartesian product of $\{(\bar{\phi}_i + 1)\frac{\pi}{2}\}_{i=1}^{\bar{m}}$, where each $\bar{\phi}_i$ is defined to be the zero of ith Legendre polynomial. More details in [22].*

When the *kernel specific* quadrature can be constructed, we always achieve exponential decrease as the integrand is a simple a trigonometric function which is analytic and its derivative can be easily bounded. On the other hand, the error characteristic of the general scheme depends on the function $p(\cot(\phi))\frac{\cos(\delta \cot(\phi))}{\sin(\phi)^2}$ for $\delta \in [0,1]$. In particular we state the following bound.

**Theorem 4** (Quadrature Fourier Features error). *Let $\Phi(x) \in \mathbb{R}^m$ be as in Definition 8 with $m = (2\bar{m})^d$, $D = [0,1]^d$, $f_\delta(\phi) = p(\cot(\phi))\frac{\cos(\delta \cot(\phi))}{\sin(\phi)^2}$ be $s - 1$ times absolutely continuous, then*

$$\sup_{x,y \in D} |k(x,y) - \Phi(x)^\top \Phi(y)| \leq d2^{(d-1)} \frac{(s+2)^{s+1}}{s!} \frac{1}{\bar{m}^{s+1}} \max_{\delta \in D} \mathrm{TV}(f_\delta^{(s)}), \tag{15}$$

*where $\mathrm{TV}$ denotes total variation on $[0,\pi]$. When $f_\delta(\phi)$ is analytic,*

$$\sup_{x,y \in D} |k(x,y) - \Phi(x)^\top \Phi(y)| \leq d2^{(d-1)} \frac{2^{2\bar{m}+1}(\bar{m}!)^4 \pi}{(2\bar{m}+1)[2\bar{m}!]^3} \max_{\delta \in [0,1], \phi \in [0,\pi]} |f_\delta^{(2m)}(\phi)|. \tag{16}$$

*Proof.* Follows from Theorems 12 and 13. □

### A.3 Modified Matérn vs Matérn kernel

In our analysis we have assumed that the Fourier transform of a kernel is decomposable (Assumption 1). Note that the Matérn kernel does not satisfy the decomposability assumption. In order to allow more generality, we define the *modified matern kernel*, which coincides with the standard Matern kernel [38] when $d = 1$, but differs for multiple dimensions. We define the modified Matérn kernel via its Fourier transform as,

$$k_n(x,y) = \int_{-\infty}^{\infty} \prod_{j=1}^d \frac{1}{(1 + \gamma^2 \omega_j^2)^n} \cos(\omega^\top (x - y)) d\omega, \qquad \text{(Modified Matérn-}n\text{)}$$

where by $d\omega = d\omega_1 \dots d\omega_j$, and $x, y \in \mathbb{R}^d$.

### A.4 Convergence of general QFF

We show the error convergence for the posterior mean of the general QFF with the modified Matern kernel in the next Figure. The convergence has been dramatically improved over RFF, although the convergence is not exponential anymore. The the trend from Theorem 4 reliably predicts the decrease.

### A.5 Proximity of quantities

In this section, we summarize theorems and lemmas that imply approximation of posterior mean and posterior variance given the uniform approximation of a kernel (Assumption 2)

**Lemma 2** (Posterior mean has bounded spectral characteristic). *Let $k$ be a kernel, $\delta \in (0, e^{-1})$ and $\{(x_t, y_t)\}_{t=1}^T$ be pairs of evaluations of function $g \in \mathcal{H}_k$, RKHS associated to $k$, then the spectral characteristic function of posterior mean $\mu_T$ is always bounded by $\frac{\|y\|_2}{\rho^2}$, and if $\|g\|_{\mathcal{H}_k} \leq B$ then the spectral characteristic function is bounded by $\frac{B\sqrt{T}}{\rho^2} + \frac{1}{\rho^2}\sqrt{(T + 2(\sqrt{T} + 1)\log(1/\delta))}$ with probability $1 - \delta$.*

Figure 4: Uniform approximation of the posterior mean for Modified Matern kernel. A sample from the corresponding GP was used, where $T$ represent the number of points in the posterior.

**Lemma 3** (Matrix error). *Let $k(x, y)$ be a kernel, and let for $\hat{k}(x, y)$ Assumption 2 be satisfied. In addition, assume that $k(x, y) \leq 1$, and $\hat{k}(x, y) \leq 1$, then this implies that for kernel matrices constructed from observations $\{x_i\}_{i=1}^t$ the following holds*

$$\left\| \mathbf{K}_t - \tilde{\mathbf{K}}_t \right\|_2 \leq \epsilon t. \tag{17}$$

*Proof.* For square matrices $\|\mathbf{A}\|_2 = \sqrt{\sum_{ij}^t |\mathbf{A}_{ij}|^2}$. As the difference between these two matrices can be writen as $\epsilon \mathbf{B}$, where $\mathbf{B}$ is symmetric matrix with elements between $-1 \leq \mathbf{B}_{ij} \leq 1$, then $\|\epsilon \mathbf{B}\|_2 = \epsilon t$. $\qquad\square$

**Lemma 4** (Proximity of spaces - existence result). *Let $k$ be a kernel defining $\mathcal{H}_k$ and $f \in \mathcal{H}_k$, its RKHS, s.t. spectral characteristic function is bounded by $B$. Assume the defining points of $f$ come from $D$. Let $\mathcal{F}_m$ approximating space with a mapping $\Phi$ s.t. this mapping defines an $\epsilon$-uniform approximation to the kernel $k$, then $\exists \hat{\mu} \in \mathcal{F}_m$ s.t. $\sup_{x \in D} |\hat{\mu}(x) - f(x)| \leq \epsilon B$.*

**Theorem 5** (Proximity of means - uniform approximation of posterior mean). *Let $k$ be a kernel defining $\mathcal{H}_k$. Let $\{(x_i, y_i)\}_{i=1}^t$ be data observation corrupted by Gaussian iid noise of $g$ satisfying Assumption 3. The variance of the noise is $\rho^2$ and $\delta \in (0, e^{-1})$. Additionally, let $\mathcal{F}_m$ be the approximating space for which proximity of spaces holds. Let the following quantities have the following definition,*

$$\mu_t = \arg \min_{\mu \in \mathcal{H}_k} \left( \sum_{j=1}^t (\mu(x_i) - y_i)^2 + \rho^2 \|\mu\|_{\mathcal{H}_k} \right) \tag{18}$$

*and*

$$\tilde{\mu}_t = \arg \min_{\mu \in \mathcal{F}_m} \left( \sum_{j=1}^t (\mu(x_i) - y_i)^2 + \rho^2 \|\mu\|_{\mathcal{F}_m} \right). \tag{19}$$

*then,*

$$\sup_{x \in D} |\mu_t(x) - \tilde{\mu}_t(x)| \leq \epsilon \frac{(t+1)^2}{\rho^2} (B + \sqrt{2 \log(1/\delta)}) \tag{20}$$

*with probability $1 - \delta$.*

**Proposition 1** (Approximation of the st. dev.). *In addition to the assumptions in Theorem 5 let $\epsilon < 1$ and $k(x, x) = 1$ then the approximated and true posterior standard deviations satisfy,*

$$\sigma_t(x) \leq \tilde{\sigma}_t(x) + \frac{2t^2 \sqrt{\epsilon}}{\rho} \tag{21}$$

*where the tilde signalizes the FF approximation.*

**Remark 1.** *The assumption that $k(x, x) = 1$ is satisfied for squared exponential kernel, and for RFF and QFF as defined in 3 and Def. 8 respectively.*

**Lemma 5** (Maximum Information gain bound for QFF). *Let $k(x, x') = \Phi(x)^\top \Phi(x)'$, where $\Phi(x)$ corresponds to the QFF mapping (5). Let $\rho$ be the magnitude of the noise variance, then, $\gamma_T$ for this kernel is smaller than $m \log(1 + \rho^{-2} T)$.*

# B Supplementary results - BO with Fourier Features

## B.1 Thompson sampling - general FF Theorem

Our analysis of Thompson sampling closely follows the analysis in [9]. In fact, we introduce the approximation into their proof technique and show that this $\epsilon$-approximation carries through the analysis and appears in the final result. The general idea is that the error in the approximation grows as $\epsilon(m)^3$ for the uncertainty estimates. If $\epsilon(m)$ decreases sufficiently quickly in $m$ - basis size, then this is not a problem. For example, for squared exponential kernel, the epsilon decreases exponentially with $m$. For modified-Matern kernels of high orders, the polynomial decrease is sufficiently fast such that this error does not dominate as well.

We are able to prove Thompson sampling with Fourier Features with a general $\epsilon$-uniform approximation.

**Theorem 6** (Thompson sampling with Fourier Features). *Let $\delta \in (0, e^{-1})$, $k$ be additive squared exponential kernel with $G$ components, and Assumptions 3 and 2 be satisfied, then running Thompson sampling with the approximated kernel where each acquisition function is optimized to the accuracy by $\alpha_t = \frac{1}{\sqrt{t}}$ suffers regret at least*

$$R_T \leq \mathcal{O}\left(\sqrt{\epsilon(m)}T^3 \log(T/\delta) + \log(T/\delta)^{3/2} \max(m \log T \sqrt{T}, G\sqrt{Tm}(\log T)^{\bar{d}/2+1})\right) \quad (22)$$

*with probability $1 - \delta$ where $\bar{d}$ is the dimension of largest additive component.*

**Corollary 1** (Same as the Theorem 3 in main text). *Under assumptions of Theorem 6, and using quadrature Fourier features from Definition (3) s.t. $m_j > 2(\log_\eta(T^3))^{d_j}$, and $m_j > \gamma_j^{-2}$ for each $j \in [G]$, the cumulative regret can be bounded*

$$R_T \leq \mathcal{O}\left(G(\log T)^{\bar{d}+1}\sqrt{T} \log\left(\frac{T}{\delta}\right)^{3/2}\right) \quad (23)$$

*with probability $1 - \delta$.*

*Proof.* Follows directly from Theorems 6 and 1. $\square$

## B.2 GP-UCB with Fourier Features

In this subsection, we present theorem bounding the regret of GP-UCB with Fourier Features, and similarly as for Thompson sampling we show that in the special case of squared exponential kernel and with a specific set of Fourier Features, the asymptotic complexity remains the same as the exact GP-UCB.

**Theorem 7.** *Let Assumptions 1, 2 and 3 be satisfied, and $\delta \in (0, e^{-1})$. The cumulative regret for a fixed horizon $T$ with the approximated kernel with $m$ Fourier Features on domain $D \subset [0, 1]^d$ using UCB acquisition function scales as*

$$\mathcal{O}\left(\beta_T^{1/2}\left(\sqrt{\epsilon(m)}\frac{T^3}{\rho^2} + \sqrt{T \log(T)^d} + \sqrt{mT}\right)\right) \quad (24)$$

*with probability $1 - 2\delta$ (the $\mathcal{O}$ hides logarithmic dependence on $\delta$ and sub-logarithmic on $T$), where $\beta_t^{1/2} = B + \sqrt{2 \log\left(\frac{1}{\delta}\right) + \gamma_t(k)}$.*

Note that the $\beta_t$ is defined for the true kernel with the maximum mutual information on the true kernel. For example, for squared exponential $\gamma_T(k) = \mathcal{O}((\log T)^{d+1})$ as in (34), and hence the regret bound,

$$R_T = \mathcal{O}\left(\sqrt{\epsilon(m)}\frac{T^3\sqrt{\log(T)^d}}{\rho^2} + \sqrt{T}\log(T)^{d+1} + \sqrt{mT \log(T)^{d+1}}\right). \quad (25)$$

**Corollary 2** (QFF with UCB). *Under the assumptions of Theorem 7, for QFF (Definition 3) and squared exponential kernel,*

$$R(T) \leq \mathcal{O}\left(\frac{T^3\sqrt{\log(T)^{d+1}}}{\eta^{m/2d}} + \sqrt{T}\log(T)^{d+1} + \sqrt{mT \log(T)^{d+1}}\right) \quad (26)$$

*with $m = 2 \log_\eta (T^3)^d$ and $\eta = (16/e)$, the regret becomes $R(T) \leq \mathcal{O}\left(\sqrt{T} \log(T)^{d+1}\right)$.*

## B.3 Details for Numerical Benchmarks

All functions have been scaled to $[-0.5, 0.5]^d$ interval which is equivalent in the convergence results as $D = [0, 1]^d$, since the diameter of the set remains the same. Each of the benchmark function is scaled by $N$ which normalizes the function such that its maximum is 1. This is done so that no additional scaling of parameters need to be introduced.

**Benchmark Functions**

$$\text{Styb-tang}(x) = -\frac{1}{2N} \sum_{i=1}^{d} (x_i/8)^4 - 16(x_i/8)^2 + 5(x_i/8) \tag{27}$$

$$\text{Michalewicz}(x) = \frac{1}{N} \sum_{i=1}^{d} \sin(x_i/\pi - 1/2) \sin\left(\frac{i(x_i - \pi/2)}{\pi^2}\right) \tag{28}$$

$$\text{Overlapping}(x) = \frac{1}{N} \sum_{i=1}^{d-1} \exp(-(x_i - x_{i+1})^2/8) \tag{29}$$

With the *Stybtang* experiment we used $\gamma = 0.6$, with *Michalewicz* $\gamma = 0.3$, with *Camelback* $\gamma = 0.1$, and with Overlapping $\gamma = 0.5$. For adversarial initialization we used initial $T$ points in the negative part of the domain. For *Michalewicz* $T = 128$ and $m = 64$ for each component, for *Stybtang* we used $m = 128$ for each component and $T = 128$, for Overlapping $T = 512$ and $m = 256$ for each component. The noise level was $\rho = 0.05$ in all experiments.

For each of the additive models we used a Lipschitz based optimization, where we specified the grid to be at most 100 points (due to computational issues). For the generalized additive model as in the *Overlapping* experiment we used L-BFGS to optimize the function iteratively on a continuous domain. In the case of Fourier Features, we use the closed form of the acquisition function.

**Free-electron laser experiment** This experiment was created by optimizing a simulator of a free electron laser. The simulator was fit with the real data from the machine. We applied normalization such that the optimum is at 1 and the noise level was considerably higher at $\rho = 1.7$ with $\gamma = 0.1$. We did not include any adversarial starting points and started without observations. The model is additive by design in the simulator so there is no model miss-match.

## C Necessary Third-party Results

**Lemma 6** (Concentration on the norm of Gaussian vector, Corollary of [27]). *Let $\zeta \sim \mathcal{N}(0, \mathbf{I}_m)$ and $\delta \in (0, e^{-1})$ then,*

$$\mathrm{P}\left(\|\zeta\|_2 \leq \sqrt{(m + 2(\sqrt{m} + 1) \log(1/\delta))}\right) \geq 1 - \delta. \tag{30}$$

### C.1 RKHS

**Theorem 8** (Corollary 3.15 in [1]). *Let $k$ be a kernel defining $\mathcal{H}_k$ and $f \in \mathcal{H}_k$, s.t $\|f\|_k \leq B$. Then with probability $(1 - \delta)$ under Gaussian noise assumption with variance $\rho^2$,*

$$|f(x) - \mu_t(x)| \leq \beta_t^{1/2} \sigma_t(x) \tag{31}$$

*for all $x$, where,*

$$\beta_t^{1/2} = B + \sqrt{2 \log\left(\frac{1}{\delta}\right) + \gamma_t(k)}. \tag{32}$$

Bounding

**Theorem 9** (Theorem 5 in [48]). *Let $D \subset \mathbb{R}^d$ be compact and convex. Assume that the kernel satisfies $k(x, x') \leq 1$.*

1. *For d-dimensional Bayesian Linear Regression*

$$\gamma_T = \mathcal{O}(d \log T). \tag{33}$$

2. *For the squared exponential kernel*

$$\gamma_T = \mathcal{O}((\log T)^{d+1}). \tag{34}$$

3. *For the Matérn kernels with $\xi > 1$*

$$\gamma_T = \mathcal{O}(T^{d(d+1)/(2\xi+d(d+1))} \log T) \tag{35}$$

Specific bounds for $\gamma_T(k)$ for additive models exist in literature. As the models are less expressive, one expects to get a smaller maximum gain values. The following theorems quantify this.

**Theorem 10** (Additive Models $\gamma_T$, Theorem 3. in [26]). *Let $k^{(1)}$ and $k^{(2)}$ be two kernel functions on the same domains $D^{(1)}$ and $D^{(2)}$. Then the additive combination $k = k^{(1)} + k^{(2)}$ defined on $D = D^{(1)} \oplus D^{(2)}$ ($\oplus$ denotes direct sum) holds that*

$$\gamma_T(k) = \gamma_T(k^{(1)}) + \gamma_T(k^{(2)}) + 2\log(T), \tag{36}$$

*where the argument signalizes the dependence on a kernel.*

**Corollary 3** (Additive squared exponential kernel). *Let $k^{(1)}$ and $k^{(2)}$ be two squared exponential kernel with inputs from dimensions $d_1$ and $d_2$ respectively. Then, for kernel $k = k^{(1)} + k^{(2)}$ is equal to $\gamma_T = \mathcal{O}\left((\log T)^{\bar{d}+1}\right)$ where $\bar{d} \overset{def}{=} \max(d_1, d_2)$.*

*Proof.* Using Theorem 10 and Equation (34), we obtain the bound. $\square$

For example, a fully additive model with the squared exponential kernel has $\gamma_T = \mathcal{O}(d(\log T)^2)$ instead of $\mathcal{O}((\log T)^{d+1})$, which shows a dramatic simplification.

## C.2 Numerical Integration

For our analysis we require standard results from Numerical Integration, namely we are require the following 3 Theorem 11, 12, and 13.

**Theorem 11** (Gauss-Hermite quadrature error, [22]). *The error on Gauss-Hermite ($w(x) = \exp(-x^2)$) quadrature with $m$ nodes for $(2m)$ times differentiable function $f$ is*

$$\epsilon(f, m) \leq \frac{m! \sqrt{\pi}}{2^m (2m)!} \max_{\epsilon \in [-\infty, \infty]} |f^{(2m)}(\epsilon)|. \tag{37}$$

**Theorem 12** (Gauss-Legengre quadrature error, [22]). *The error on Gauss-Lgengre ($w(x) = 1$) quadrature with $m$ nodes for $(2m)$ times differentiable function $f$ is*

$$\epsilon(f, m) \leq \frac{2^{2m+1}(m!)^4}{(2m+1)[2m!]^3} \max_{\epsilon \in [-\infty, \infty]} |f^{(2m)}(\epsilon)|. \tag{38}$$

**Theorem 13** (Theorem 1 in [11]). *Let $Q_n$ be a positive interpolatory quadrature formula with $n$ nodes for the weight $w$, and assume $w(x) \leq M(1x^2)^{-1/2}$ for some $M > 0$. Then,*

$$|I(wf) - Q(f)|_\infty \leq 5M((s+2)\pi)^{s+1}(s!)^{-1}n^{-s-1} \text{TV}(f)(s)$$

*whenever the total variation is finite, for some $s \in \mathbb{N}_0$.*

# D  Proofs of the Results

## D.1  From Univariate to Multivariate Integration

In standard textbooks, numerical integration is usually formulated for functions defined on 1D real interval. Here, we prove necessary lemmas and finally a proposition, which extends the convergence properties to Cartesian product grids.

**Lemma 7.** *Let us suppose we have two tuples of positive real numbers $(a_1, \ldots, a_d)$ and $(\hat{a}_1, \ldots, \hat{a}_d)$ s.t. $|a_i - \hat{a}_i| \leq \epsilon_i$, and each $|a_i|$ and $|\hat{a}_i|$ are bounded by a positive constant $p$, then*

$$\left| \prod_{j=1}^{d} a_j - \prod_{j=1}^{d} \hat{a}_j \right| \leq p^{(d-1)} \sum_{j=1}^{d} \epsilon_j. \tag{39}$$

*Proof.* We first prove the assertion for $d = 2$. There are four combinations of possible orderings of $a_1, a_2$ and $\hat{a}_1, \hat{a}_2$. However, the problem is symmetric, as a consequence one combination suffices to show the lemma. Suppose $a_1 \geq \hat{a}_1$, and $a_2 \geq \hat{a}_2$, then $\hat{a}_1 \leq a_1 \leq \hat{a}_1 + \epsilon_1$ and $\hat{a}_2 \leq a_2 \leq \hat{a}_2 + \epsilon_2$. Multiplying the first by $a_2$ and the second by $\hat{a}_1$, we obtain.

$$\hat{a}_1 a_2 \leq \quad a_1 a_2 \quad \leq \hat{a}_1 a_2 + \epsilon_1 a_2 \tag{40}$$
$$\hat{a}_2 \hat{a}_1 \leq \quad a_2 \hat{a}_1 \quad \leq \hat{a}_2 \hat{a}_1 + \epsilon_2 \hat{a}_1 \tag{41}$$

Then

$$a_1 a_2 - \hat{a}_1 \hat{a}_2 \leq \hat{a}_1 a_2 + \epsilon_1 a_2 - \hat{a}_1 \hat{a}_2 \leq \epsilon_1 a_2 + \epsilon_2 \hat{a}_1 \leq 2p(\epsilon_1 + \epsilon_2)$$

and

$$a_1 a_2 - \hat{a}_1 \hat{a}_2 \geq \hat{a}_1 a_2 - \hat{a}_1 \hat{a}_2 \geq 0.$$

which completes the proof for $d = 2$.

Using mathematical induction, we assume the result is valid for $d$ and prove that the result has to hold for $d + 1$. Namely, assume shorthand $b = \prod_{j=1}^{d} a_j$ and $\hat{b} = \prod_{j=1}^{d} \hat{a}_j$, then

$$\left| b - \hat{b} \right| \leq p^{(d-1)} \sum_{j=1}^{d} \epsilon_j = \varepsilon,$$

and both $|b|$ and $|\hat{b}|$ are bounded by $p^k$. Thus, we can proceed in the same manner as we did for $d = 2$, only in the last step we take different error and bound, to arrive at,

$$b a_{d+1} - \hat{b} \hat{a}_{d+1} \leq p\varepsilon + p^{d-1} \epsilon_{d+1} \leq p^d \sum_{j=1}^{d} \epsilon_j + p^d \epsilon_{d+1},$$

which leads to

$$\left| b a_{d+1} - \hat{b} \hat{a}_{d+1} \right| \leq p^d \sum_{j=1}^{d+1} \epsilon_j.$$

The positivity of the difference is ensured using a similar argument as for $d = 2$.  $\square$

**Proposition 2.** *Let $\omega \in \mathbb{R}^d$, and $\delta \in [0, 1]^d$. Under Assumption 1, assume that a quadrature rule of each single integral $\int p_i(\omega_i) \cos(\omega_i \delta_i) d\omega_i$ or $\int p(\omega_i) \sin(\omega_i \delta_i) d\omega_i$ can be upper bounded with $\epsilon$ accuracy for each coordinate $i$, then the quadrature error of $\int p(\omega) \cos(\omega^\top \delta) d\omega$ scales as $d 2^{(d-1)} \epsilon$.*

*Proof of Proposition 2.*

$$I = \int \prod_{i=1}^{d} p_i(\omega_i) \cos(\omega^\top x) d\omega \tag{42}$$

$$= \int p_1(\omega_1) \cos(\omega_1^\top x_1) d\omega_1 \int \prod_{i=2}^{d} p(\omega_i) \cos(\omega_2 x_2 + \ldots) d\omega_2 \ldots d\omega_d \tag{43}$$

$$-\int p(\omega_1)\sin(\omega_1^\top x_1)d\omega_1 \int \prod_{i=2}^{d} p(\omega_i)\sin(\omega_2 x_2 + \dots)d\omega_2 \dots d\omega_d \tag{44}$$

$$= \quad \dots \tag{45}$$

Applying this, we notice that we have $2^{d-1}$ terms in our decomposition where each combination of $\sin$ and $\cos$ with $\omega_1 \dots \omega_d$ appears once. Each term is product of $d$ integrals which can be approximated with an error $\epsilon$.

Using Lemma 7 and the fact that the integrals are bounded by 1, we can bound the quadrature error of the whole expression as $d2^{d-1}\epsilon$. □

## D.2 Quadrature Fourier Features - QFF

*Proof of theorem 1.* We first show that we get the desired decrease for $d = 1$, and subsequently generalize using the Proposition 2.

The integral we are trying to approximate

$$k(\delta) = \int_{-\infty}^{\infty} \exp\left(\frac{\gamma^2\omega^2}{2}\right)\cos(\omega\delta)d\omega \tag{46}$$

can be rewritten using $x = \omega\gamma/2$ to get

$$k(\delta) = \frac{\sqrt{2}}{\gamma}\int_{-\infty}^{\infty}\exp(-x^2)\cos\left(\frac{\delta\sqrt{2}}{\gamma}x\right). \tag{47}$$

Now, we observe that the weighting function is $\exp(-x^2)$ for standard Gauss-Hermite quadrature with integrand being $f(x) = \cos(\frac{\delta\sqrt{2}}{\gamma}x)$. In order to apply Theorem 11, we require the maximum of $|f^{(2\bar{m})}(x)|$,

$$\max_{x\in[-\infty,\infty]}|f^{(2\bar{m})}(x)| = \left(\frac{\delta\sqrt{2}}{\gamma}\right)^{2m} \overset{\delta\leq 1}{\leq} \left(\frac{\sqrt{2}}{\gamma}\right)^{2\bar{m}}. \tag{48}$$

Using this expression and Theorem 11, we arrive at, where $d = 1$

$$\sup_{x,y\in D}|k(x,y) - \Phi(x)^\top\Phi(y)| \leq \frac{\bar{m}!\sqrt{\pi}}{2^{\bar{m}}(2\bar{m})!}\left(\frac{\sqrt{2}}{\gamma}\right)^{2\bar{m}}. \tag{49}$$

Using Stirling approximation, we can make the estimate factorial free,

$$\sup_{x,y\in D}|k(x,y) - \Phi(x)^\top\Phi(y)| \leq \sqrt{\frac{\pi}{2}}\frac{1}{\bar{m}^{\bar{m}}}\left(\frac{e}{4\gamma^2}\right)^{\bar{m}} \tag{50}$$

Using the fact that we have tensor product grid with Assumption 1 and Proposition 2 the result generalizes with the proper scaling factor. □

## D.3 Proximity of mean and variance

*Proof of Lemma 2.* Due to the Representer theorem [43], one can express $\mu_T(x) = \sum_{t=1}^{T}\alpha_t k(x, x_t) = \alpha_T^\top k_T(x)$, where in the regularized RKHS estimate $\alpha_T = (\mathbf{K} + \rho^2\mathbf{I})^{-1}y$, where evaluation $\{y\}_{t=1}^{T}$ are stacked in a vector and the matrix $\mathbf{K} = k(x_i, x_j)$ for $i, j \in [T]$.

Using the definition of spectral characteristic function $\alpha(\omega) = \sum_{j=1}^{t}\alpha_t\exp(-i\omega x_t)$, and $|\alpha(\omega)| = \sqrt{\|\alpha_T\|^2} = \sqrt{y^\top(\mathbf{K} + \mathbf{I}\rho^2)^{-2}y} \leq \frac{1}{\rho^2}\|y\|$. As $y$ is finite, the spectral characteristic function must be bounded as well. Further, each $y_t = g(x_t) + \epsilon_t$ and given that $|g(x)| \leq B$. Consequently, we are able to provide a concentration result using Lemma 6. Namely, we have that $P(\|y\|_2 \leq \sqrt{T}B + \sqrt{(T + 2(\sqrt{T} + 1)\log(1/\delta))}) \geq 1 - \delta$. Multiplying by $\rho^{-2}$ finishes the proof. □

*Proof of Lemma 1.* Complex conjugate is denoted as a star ($*$).

$$|\alpha(\omega)|^2 = \left(\sum_{j\in I}\alpha_j\exp(i\omega^\top x_j)\right)^* \left(\sum_{j\in I}\alpha_j\exp(i\omega^\top x_j)\right) = \left(\sum_{i,j\in I}\alpha_j\alpha_i\exp(i\omega^\top(x_j-x_i))\right).$$

Also note that

$$\|g\|_k^2 = \sum_{ij\in I}\alpha_i\alpha_j k(x_i-x_j) = \int_{\mathbb{R}^d}p(\omega)\sum_{ij\in I}\alpha_i\alpha_j\exp(i\omega^\top(x_i-x_j))d\omega = \mathbb{E}_{p(\omega)}[|\alpha(\omega)|^2].$$

Now, $\|g\|_k^2 = \mathbb{E}_{p(\omega)}[|\alpha(\omega)|^2] \leq \max_\omega |\alpha(\omega)|^2 \int_{\mathbb{R}^d}p(\omega)d\omega \leq B$, as $p(\omega)$ is a probability distribution. $\qquad\square$

*Proof of Lemma 4.* The proof goes by construction. We know that $f = \sum_{j\in I}k(x,x_j)\alpha_i$, where $I$ is possibly countably infinite index set. The spectral characteristic function $\alpha(\omega) \overset{\text{def}}{=} \sum_{j\in I}\alpha_j\exp(i\omega^\top x_j)$ is bounded with B by assumption. We construct the following $\hat{\mu} = \sum_{j\in I}\alpha_j\Phi(x)^\top\Phi(x_j)$, and show the following relation for it

$$\sup_{x\in D}|\hat{\mu}(x)-f(x)| \leq \sum_{j\in I}\alpha_j\sup_{x\in D}|\Phi(x)^\top\Phi(x_j)-k(x,x_j)| \tag{51}$$

$$\leq \max_\omega|\alpha(\omega)|\epsilon \leq B\epsilon, \tag{52}$$

which finishes the proof. $\qquad\square$

*Proof of Theorem 5.* Let us define $\hat{\mu}_t$ be defined as in Lemma 4. In general the quantities with tilde come from the approximated kernel. We use the shorthand for vector $\tilde{k}\in\mathbb{R}^t$ s.t. $\tilde{k}_t(x)_j = \tilde{k}(x_j,x)$, and similar for the kernel matrix $\mathbf{K}_t$ at time t. We insert this quantity into the following chain of reasoning,

$$\sup_{x\in D}|\mu_t(x)-\tilde{\mu}_t(x)| \leq \sup_{x\in D}|\mu_t(x)-\hat{\mu}_t(x)| + \sup_{x\in D}|\hat{\mu}_t(x)-\tilde{\mu}_t(x)|$$

$$\overset{\text{Lem. 4}}{\leq} \epsilon B + \sup_{x\in D}|\hat{\mu}_t(x)-\tilde{\mu}_t(x)|$$

$$\overset{\text{Representer thm.}}{=} \epsilon B + \sup_{x\in D}|\tilde{k}_t(x)^\top(\alpha-\tilde{\alpha})|$$

$$\overset{\text{by def.}}{\leq} \epsilon B + \|\alpha-\tilde{\alpha}\|_1$$

$$\leq \epsilon B + \left\|((\mathbf{K}_t+\rho\mathbf{I})^{-1}-(\tilde{\mathbf{K}}_t+\rho\mathbf{I})^{-1})y\right\|_1$$

$$\overset{\text{norm. eq.}}{\leq} \epsilon B + \sqrt{t}\left\|((\mathbf{K}_t+\rho^2\mathbf{I})^{-1}-(\tilde{\mathbf{K}}_t+\rho^2\mathbf{I})^{-1})y\right\|_2$$

$$\leq \epsilon B + \sqrt{t}\left\|(\mathbf{K}_t+\rho^2\mathbf{I})^{-1}-(\tilde{\mathbf{K}}_t+\rho^2\mathbf{I})^{-1}\right\|_2\|y\|_2$$

$$\overset{\text{submultiplicative}}{\leq} \epsilon B + \sqrt{t}\left\|(\tilde{\mathbf{K}}_t+\rho^2\mathbf{I})^{-1}\right\|_2\left\|(\mathbf{K}_t+\rho^2\mathbf{I})^{-1}\right\|_2\left\|(\mathbf{K}_t-\tilde{\mathbf{K}}_t)\right\|_2\|y\|_2$$

$$\overset{\text{Bounded}}{\leq} \epsilon B + \frac{\sqrt{t}}{\rho^2}\left\|(\mathbf{K}_t-\tilde{\mathbf{K}}_t)\right\|_2\|y\|_2$$

$$\overset{\text{Bounded}}{\leq} \epsilon B + \frac{\sqrt{t}}{\rho^2}\left\|(\mathbf{K}_t-\tilde{\mathbf{K}}_t)\right\|_2\|y\|_2$$

$$\overset{\text{(17), Lemma 2}}{\leq} \epsilon\left(\frac{\|y\|_2}{\rho^2}+\frac{t^{3/2}}{\rho^2}\|y\|_2\right)$$

$$= \epsilon\frac{\|y\|_2}{\rho^2}\left(1+t^{3/2}\right)$$

$$\overset{\text{Lemma6}}{\leq} \frac{\epsilon(t^{3/2}+1)}{\rho^2}\left(\sqrt{t}B+\sqrt{(t+2(\sqrt{t}+1)\log(1/\delta))}\right)$$

where the last step holds with probability $1 - \delta$. We choose to provide a harsher but less technical upper bound,

$$\sup_{x \in D} |\mu_t(x) - \tilde{\mu}_t(x)| \le \epsilon \frac{(t+1)^2}{\rho^2}(B + \sqrt{2\log(1/\delta)}) \tag{53}$$

with $1 - \delta$ which finishes the proof. $\square$

*Proof of Proposition 1.* We employ a similar technique as in the proof of the Theorem 5. Let $v(x) \in \mathbb{R}^t$ be vector with bounded elements $|v_i| \le 1$ s.t.

$$\tilde{k}_t(x) = k_t(x) + \epsilon v(x), \tag{54}$$

where $\epsilon$ comes from $\epsilon$-uniform approximation of the kernel.

$$
\begin{aligned}
\sigma_t(x)^2 &= 1 - k_t(x)^\top (\mathbf{K}_t + \rho^2 \mathbf{I})^{-1} k_t(x) \\
&= \tilde{\sigma}_t(x)^2 - k_t(x)^\top (\mathbf{K}_t + \rho^2 \mathbf{I})^{-1} k_t(x) + \tilde{k}_t(x)^\top (\tilde{\mathbf{K}}_t + \rho^2 \mathbf{I})^{-1} \tilde{k}_t(x) \\
&\stackrel{(54)}{=} \tilde{\sigma}_t(x)^2 + k_t(x)^\top (-(\mathbf{K}_t + \rho^2 \mathbf{I})^{-1} + (\tilde{\mathbf{K}}_t + \rho^2 \mathbf{I})^{-1}) k_t(x) + 2\epsilon v(x)^\top (\tilde{\mathbf{K}}_t + \rho^2 \mathbf{I})^{-1} k_t(x) \\
&\quad + \epsilon^2 v(x)^\top (\tilde{\mathbf{K}}_t + \rho^2 \mathbf{I})^{-1} v(x) \\
&\stackrel{\text{pos. def.}}{\le} \tilde{\sigma}_t(x)^2 + k_t(x)^\top ((\mathbf{K}_t + \rho^2 \mathbf{I})^{-1} - (\tilde{\mathbf{K}}_t + \rho^2 \mathbf{I})^{-1}) k_t(x) + 2\epsilon v(x)^\top (\tilde{\mathbf{K}}_t + \rho^2 \mathbf{I})^{-1} k_t(x) + \frac{\epsilon^2 t}{\rho^2} \\
&\le \tilde{\sigma}_t(x)^2 + t \frac{k_t(x)^\top ((\mathbf{K}_t + \rho^2 \mathbf{I})^{-1} - (\tilde{\mathbf{K}}_t + \rho^2 \mathbf{I})^{-1}) k_t(x)}{k_t(x)^\top k_t(x)} + 2\epsilon v(x)^\top (\tilde{\mathbf{K}}_t + \rho^2 \mathbf{I})^{-1} k_t(x) + \frac{\epsilon^2 t}{\rho^2} \\
&\le \tilde{\sigma}_t(x)^2 + t \lambda_{max}((\mathbf{K}_t + \rho^2 \mathbf{I})^{-1} - (\tilde{\mathbf{K}}_t + \rho^2 \mathbf{I})^{-1}) + \frac{2t^2 \epsilon}{\rho^2} + \frac{\epsilon^2 t}{\rho^2} \\
&= \tilde{\sigma}_t(x)^2 + t \left\| (\mathbf{K}_t + \rho^2 \mathbf{I})^{-1} - (\tilde{\mathbf{K}}_t + \rho^2 \mathbf{I})^{-1} \right\|_2^2 + \frac{2t^2 \epsilon}{\rho} + \frac{\epsilon^2 t}{\rho^2} \\
&\le \tilde{\sigma}_t(x)^2 + \frac{t}{\rho^2} \left\| \mathbf{K}_t - \tilde{\mathbf{K}}_t \right\|_2^2 + \frac{2t^2 \epsilon}{\rho^2} + \frac{\epsilon^2 t}{\rho^2} \\
&\stackrel{(17)}{\le} \tilde{\sigma}_t(x)^2 + \frac{t^3 \epsilon^2}{\rho^2} + \frac{2t^2 \epsilon}{\rho^2} + \frac{\epsilon^2 t}{\rho^2} \\
&\le \tilde{\sigma}_t(x)^2 + \frac{4t^3 \epsilon}{\rho^2}
\end{aligned}
$$

Again we use a harsh bound, but for the final results it does not matter, Hence, $\sigma_t(x) \le \tilde{\sigma}_t(x) + \frac{2t^{3/2}\sqrt{\epsilon}}{\rho} \le \tilde{\sigma}_t(x) + \frac{2t^2 \sqrt{\epsilon}}{\rho}$ due to sub-additivity. $\square$

### D.4 Polynomial algorithm

*Proof of Lemma 6.* As the norm of a multivariate vector of dimension $m$ is distributed with $\chi^2_m$ distribution to get the tail bound we use a appropriate tail bound. Namely, we use a result from [27] that $P(\|\zeta\|_2^2 \ge m + \sqrt{2mx} + 2x) \le \exp(-x)$, which yields after simplification that

$$P(\|\zeta\|_2^2 \le m + 2(\sqrt{m} + 1)\log(1/\delta)) \ge 1 - \delta, \tag{55}$$

for $\delta \in (0, e^{-1})$. Taking the square root of the result, which is monotone transformation yields the result. This finishes the proof. $\square$

*Proof of Lemma 5.* Note first that $\Phi(\mathbf{X}_T) \in \mathbb{R}^{T \times m}$. Then,

$$
\begin{aligned}
\gamma_T &\le \log \left| \mathbf{I} + \rho^{-2} \Phi(\mathbf{X}_T) \Phi(\mathbf{X}_T)^\top \right| \\
&= \log \left| \mathbf{I} + \rho^{-2} \Phi(\mathbf{X}_T)^\top \Phi(\mathbf{X}_T) \right| \\
&\stackrel{\text{Hadamard ineq.}}{\le} \log \left| \mathbf{I} + \rho^{-2} \mathbf{D} \right| \\
&\le m \log \left| 1 + \rho^{-2} \lambda_{max}(\mathbf{D}) \right|
\end{aligned}
$$

$$\leq \qquad m \log \left| 1 + \rho^{-2} T \right|$$

where $\mathbf{D} = \mathbf{I}_{m \times m} \operatorname{diag}(\Phi(\mathbf{X}_T)^\top \Phi(\mathbf{X}_T))$, which are by definition $T$, as each $\|\Phi(x)\|_2 = 1$ due to Remark 1. $\qquad \square$

**Lemma 8.** *Let $k$ be an approximated kernel with Fourier features, $k(x, y) = \Phi(x)^\top \Phi(y)$, where $\Phi(x) \in \mathbb{R}^m$, and $\Phi(x)_j = v_j \cos(\omega_j^\top x)$ and $\Phi(x)_{j+m/2} = v_j \sin(\omega_j^\top x)$ for $j \in [0, m/2]$. In addition, let the following quantities be defined as follows,*

$$\mathbf{\Sigma} \stackrel{def}{=} \Phi(\mathbf{X})^\top \Phi(\mathbf{X}) + \rho^2 \mathbf{I} \tag{56}$$

$$\nu \stackrel{def}{=} (\mathbf{\Sigma})^{-1} \Phi(\mathbf{X})^\top y, \tag{57}$$

*then the sample from posterior $f$ given $(\mathbf{X}, y)$ has form $f(x) = \Phi(x)^\top \theta$, where $\theta \sim \mathcal{N}(\nu, \mathbf{\Sigma}^{-1})$. Moreover, the sample is Lipschitz continuous with $L = \|\theta\|_2 \sqrt{\sum_{i=1}^{m/2} 2v_i^2 \|\omega_i\|^2}$ with respect to Euclidean norm.*

*Proof.* The first statement follows from considerations in [56]. For the second claim we show $L$ is suitable constant s.t. the following holds

$$|\theta^\top (\Phi(x) - \Phi(y))| \leq L \|x - y\|_2. \tag{58}$$

Trigonometric function $\cos$ and $\sin$ are trivially 1-Lipschitz continuous due to mean value theorem. Thus,

$$
\begin{aligned}
\|\Phi(x) - \Phi(y)\|_2 &= \sqrt{\sum_{i=1}^{m/2} v_i^2 ((\cos(\omega_i^\top x) - \cos(\omega_i^\top y))^2 + (\sin(\omega_i^\top x) - \sin(\omega_i^\top y))^2)} \\
&\leq \sqrt{\sum_{i=1}^{m/2} 2v_i^2 \|\omega_i\|^2 \|x - y\|_2^2} = \sqrt{\sum_{i=1}^{m/2} 2v_i^2 \|\omega_i\|^2} \|x - y\|_2,
\end{aligned}
$$

which can be used to provide a bound on a Cauchy-Schwartz of (58). This completes the proof. $\qquad \square$

**Lemma 9** (Iterative updates). *Let the quantities in (57) and (56) be defined over $\mathbf{X}_t$, then the following holds two iterative updates rules for the two quantities defined over $\mathbf{X}_{t+1}$, which is obtained by contacatenaing observation $x_{t+1}$ to the matrix, are as follows,*

$$\nu_{t+1} = \nu_t - \frac{(\mathbf{\Sigma}_t)^{-1} \Phi(x_{t+1})(\Phi(x_{t+1})^\top \nu_t - y_{t+1})}{1 + \Phi(x_{t+1})^\top (\mathbf{\Sigma}_t)^{-1} \Phi(x_{t+1})} \tag{59}$$

$$\mathbf{\Sigma}_{t+1} = \mathbf{\Sigma}_t + \Phi(x_{t+1}) \Phi(x_{t+1})^\top. \tag{60}$$

*Proof.* The second equation is trivial and the first one can be obtained by applying Sherman-Woodbury rank one inverse and assembling the relation. $\qquad \square$

**Lemma 10** (Bounded mean norm). *Let the quantities in (57) be defined over $\mathbf{X}_t$, then we can present a following bound on the approximated mean,*

$$\|\nu_T\|_2 \leq \sqrt{\beta_T \gamma_T(k) T} + T \sqrt{2\rho^2 \log(3/\delta)} + \epsilon \frac{(T+1)^3}{3\rho^2} (B + \sqrt{2 \log(3/\delta)}) \tag{61}$$

*with probability $1 - \delta$ and $\|\nu_t\|_2 = 0$*

*Proof.* Let us define a shorthand $C_{t-1} = \Phi(x_t)^\top \nu_{t-1} - y_t$. Moreover, Lemma 9 implies a Löwner ordering on the matrices

$$\mathbf{\Sigma}_t \succeq \mathbf{\Sigma}_{t-1} \succeq \mathbf{I}\rho^2. \tag{62}$$

Hence, using the iterative update from Lemma 9 we know

$$\|\nu_t\|_2 \quad \leq \quad \|\nu_{t-1}\|_2 + |C_{t-1}| \frac{\sqrt{\Phi(x_t)^\top (\mathbf{\Sigma}_{t-1})^{-2} \Phi(x_t)}}{1 + \Phi(x_t)^\top (\mathbf{\Sigma}_{t-1})^{-1} \Phi(x_t)}$$

$$\overset{(62)}{\leq} \quad \|\nu_{t-1}\|_2 + |C_{t-1}|\rho \, \|\Phi(x_t)\|_2$$

$$\overset{\|\Phi(x)\|_2 = 1}{\leq} \quad \|\nu_{t-1}\|_2 + |C_{t-1}|\rho$$

Now, $|C_{t-1}| \leq |\tilde{\mu}_{t-1}(x_t) - g(x_t) - \epsilon_t| \leq |\tilde{\mu}_{t-1}(x_t) - \mu_{t-1}(x_t)| + \beta_t^{1/2}\sigma(x_t) + \sqrt{2\rho^2 \log(2/\delta)}$ with probability $1 - \delta$. Using Theorem 5,

$$|C_{t-1}| \leq \epsilon \frac{(t)^{3/2}}{\rho^2}(B + \sqrt{2\log(2/\delta)}) + \beta_t^{1/2}(\delta/3)\sigma(x_t) + \sqrt{2\rho^2 \log(3/\delta)}$$

with probability $1 - \delta$ (we used union bound and suitably scaled the $\delta$, i.e. $\beta_t$ is also function of $\delta$), where $A = \|\mu_t\|_{\mathcal{H}_k}$.

Substituting into the above equation and summing the recurrence,

$$\|\nu_T\|_2 \leq \sqrt{\beta_T \gamma_T(k)T} + T\sqrt{2\rho^2 \log(3/\delta)} + \epsilon \frac{(T)^3}{3\rho^2}(B + \sqrt{2\log(3/\delta)}) \tag{63}$$

with probability $1 - \delta$. $\qquad \square$

**Theorem 14** (Polynomial number of steps). *Let $\delta \in (0, e^{-1})$ and $k$ be an additive non-overlapping kernel with $G$ groups and $\bar{d} = \max_{j \in G} d_j$ - maximal dimension. Moreover, let $\Phi(\cdot) \in \mathbb{R}^{m_j}$ be the approximation of the additive component $j$ of a kernel as in Definition 3, where $m_j \geq 2 \log_\eta (T^3)^{d_j}$ and $m_j \geq \frac{1}{\gamma_j^2}$ where $\eta = 16/e$. Running the Algorithm 1 with any global optimization with accuracy $\alpha$ requires at most*

$$\mathcal{O}\left(\frac{G \log(\frac{T}{\delta})^{\bar{d}/2}}{\alpha^{\bar{d}}}\left(T^{3/2}(\log T)^{\bar{d}} + T^2 (\log T)^2\right)^{\bar{d}}\right) \tag{64}$$

*evaluation of black-box function with probability $1 - \delta$.*

*Proof.* As the algorithm iterates on the group, we can bound the number of evaluations done by looking at the largest group $d_j$ and assuming that for each group the same number of evaluations needs to be done. Let $M$ be number of Fourier Features basis of the largest group, namely according to the definition of we have $M = 2 \log(T^3)^{d_j}$.

Due to Theorem 1 consequently becomes (using a harsh-bound),

$$\epsilon(T) \leq G\sqrt{\frac{\pi}{2}}T^{-3} \tag{65}$$

Any global optimization of Lipschitz continuous functions solving the optimization to the accuracy $\alpha$ requires at most

$$G\left(\frac{L}{\alpha} + 1\right)^{\bar{d}} \tag{66}$$

evaluations of the black-box function [34].

According to the Lemma 8 the acquisition function we need to optimize at each round is Lipschitz continuous with $L_t = \|\theta_t\|_2 \sqrt{\sum_{i=1}^{M/2} 2v_i^2 \|\omega_i\|^2}$. Using the decomposition

$$\|\theta_t\|_2 = \left\|\nu_t + \Sigma_t^{1/2}\zeta_t\right\| \leq \|\nu_t\| + \frac{1}{\rho}\|\zeta\|,$$

where $\zeta \sim \mathcal{N}(0, \mathbf{I}_M)$ (Consequence of (62)).

In order to obtain a concentration on $L_t$, we use Lemma 6 with $M$ dimensional vectors,

$$\mathrm{P}\left(\|\theta_t\| \leq (\|\nu_t\| + \rho^{-1}\sqrt{(M + 2(\sqrt{M} + 1)\log(1/\delta))})\right) \geq 1 - \delta,$$

and hence consequently,

$$P\left(L_t \leq \sqrt{\sum_{i=1}^{m/2} 2v_i^2 \|\omega_i\|^2}(\|\nu_t\| + \rho^{-1}\sqrt{(M + 2(\sqrt{M}+1)\log(T/\delta))}) \text{ for all } t \leq T\right) \geq 1 - \delta.$$

Substituting for $\|\nu_t\|$ with twice the $\delta$ from (61), specifies the concentration inequality fully.

To upper-bound the number of evaluation we insert $L_t$ into (66), and sum over the finite horizon $T$

$$\frac{G}{\alpha^{\bar{d}}} \sum_{t=1}^{T} \left(\bar{S}\left(\|\nu_t\| + \rho^{-1}\sqrt{(M + 2(\sqrt{M}+1)\log(2T/\delta))}\right) + \alpha\right)^{\bar{d}} \tag{67}$$

where $\bar{S} = \sqrt{\sum_{i=1}^{M/2} 2v_i^2 \|\omega_i\|^2}$. Using asymptotic results from (61), and with (34) and the max bound of the sum, we get

$$\mathcal{O}\left(\frac{G \log(\frac{T}{\delta})^{\bar{d}/2}}{\alpha^{\bar{d}}}\left(T^{3/2}(\log T)^{\bar{d}+1} + T^2(\log T)^2\right)^{\bar{d}}\right) \tag{68}$$

with probability 1-$\delta$. $\qquad\square$

**Corollary 4.** *Under the assumptions of Theorem 14 assuming fully additive non-overlapping the number of black-box evaluations in Algorithm 1 is at most,*

$$\mathcal{O}\left(\frac{d\sqrt{\log(T/\delta)}}{\alpha}\left(T^2(\log T)^2\right)\right).$$

### D.5 UCB

*Proof of Theorem 7.*

$$
\begin{aligned}
R_T &= \sum_{t=1}^{T} g(x^*) - g(x^t) \\
&\overset{(31)}{\leq} \sum_{t=1}^{T} \mu_t(x^*) + \beta_t^{1/2}\sigma_t(x^*) - g(x^t) \\
&\leq \sum_{t=1}^{T} \tilde{\mu}_t(x^*) + \epsilon\frac{(t+1)^{3/2}}{\rho^2}(B + \sqrt{2\log(1/\delta)}) + \beta_t^{1/2}\sigma_t(x^*) - g(x^t) \\
&\leq \sum_{t=1}^{T} \tilde{\mu}_t(x^*) + \epsilon\frac{(t+1)^{3/2}}{\rho^2}(B + \sqrt{2\log(1/\delta)}) + \beta_t^{1/2}\left(\tilde{\sigma}_t(x^*) + \frac{\epsilon t^2}{\rho^2}\right) - g(x^t) \\
&\overset{\text{alg.}}{\leq} \sum_{t=1}^{T} \tilde{\mu}_t(x^t) + \epsilon\frac{(t+1)^{3/2}}{\rho^2}(B + \sqrt{2\log(1/\delta)}) + \beta_t^{1/2}\left(\tilde{\sigma}_t(x^t) + \frac{\sqrt{\epsilon}t^2}{\rho^2}\right) - g(x^t) \\
&\leq \sum_{t=1}^{T} \mu_t(x^t) + 2\epsilon B\frac{(t+1)^{3/2}}{\rho^2}(B + \sqrt{2\log(1/\delta)}) + \beta_t^{1/2}\left(\tilde{\sigma}_t(x^t) + \frac{\sqrt{\epsilon}t^2}{\rho^2}\right) - g(x^t) \\
&\leq \sum_{t=1}^{T} 2\epsilon B\frac{(t+1)^{3/2}}{\rho^2}(B + \sqrt{2\log(1/\delta)}) + \beta_t^{1/2}\left(\tilde{\sigma}_t(x^t) + \frac{\sqrt{\epsilon}t^2}{\rho^2}\right) + \beta_t^{1/2}\sigma_t(x^t) \\
&\leq 2\epsilon B\frac{(T)^3}{\rho^2}(B + \sqrt{2\log(1/\delta)}) + \beta_T^{1/2}\sum_{t=1}^{T}\left(\tilde{\sigma}_t(x^t) + \sigma_t(x^t)\right) + \frac{\sqrt{\epsilon}t^2}{\rho^2} \\
&\leq 2\epsilon B\frac{(T)^3}{\rho^2}(B + \sqrt{2\log(1/\delta)}) + \beta_T^{1/2}\left(C_1\sqrt{T\log(T)^d} + C_2\sqrt{mT}\right) + \frac{\sqrt{\epsilon}T^3\beta_T^{1/2}}{3\rho^2}
\end{aligned}
$$

$$= \mathcal{O}\left(\beta_T^{1/2}\left(\sqrt{\epsilon(m)}\frac{T^3}{\rho^2} + \sqrt{T\log(T)^d} + \sqrt{mT})\right)\right)$$

We use Theorem 8 twice, holds with probability $1-\delta$. Similarly, for each $t$ we use twice the theorem on proximity of means which holds with $1-\delta$. Taking union bound over these, gives us that the final inequality holds with $1-2\delta$ where the probability enters through in $\beta_T$. $\square$

### D.6 Thompson sampling - I

**Definition 9** (Necessary definitions). *We work we a filtration $\mathfrak{F}_{t-1}$, which incorporates all events that lead up until the sampling of $\{(x_j, y_j)\}_{j=1}^{t-1}$.*

$$\beta_t^{1/2} \overset{def}{=} B + \frac{1}{\rho}\sqrt{2\log(1/\delta) + \gamma_t(k)} \tag{69}$$

$$\tilde{\beta}_t^{1/2} \overset{def}{=} \frac{1}{\rho}\sqrt{2\log(1/\delta) + m} \tag{70}$$

$$B_t^{1/2} \overset{def}{=} \max(\beta_t^{1/2}, \tilde{\beta}_t^{1/2}) \tag{71}$$

$$\xi(\epsilon, t) \overset{def}{=} \sqrt{\epsilon}\left(\frac{t^2}{\rho^2}(B + \sqrt{2\log(1/\delta_y)} + 4\rho B_t^{1/2})\right) \tag{72}$$

*Let the set of saturated arms be,*

$$S_t \overset{def}{=} \{x \in D | g(x^*) - g(x) \le 2B_t^{1/2}\tilde{\sigma}_{t-1}(x) + \xi(\epsilon, t)\} \tag{73}$$

**Lemma 11** (Concentration for a linear sample path). *Let $\theta_t \sim \mathcal{N}(\nu_{t-1}, (\Sigma_{t-1})^{-1})$, then let sample of approximated posterior be $\tilde{f}_t(x) = \Phi(x)^\top \theta_t$, where $\Phi(x) \in \mathbb{R}^{2m}$ is as in the Definition 3, then*

$$|\tilde{f}_t(x) - \tilde{\mu}_{t-1}(x)| \le \tilde{\beta}_t^{1/2}\tilde{\sigma}_t(x) \tag{74}$$

*with probability $1-\delta$.*

*Proof.* We can introduce the following decomposition $\theta_t = \nu_{t-1} + \Sigma_{t-1}^{-1/2}\varepsilon$, where $\varepsilon \sim \mathcal{N}(0, \mathbf{I}_m)$. Using decomposition of Fourier features $\tilde{\mu}_{t-1} = \nu_{t-1}^\top\Phi(x)$, $|\tilde{f}_t(x) - \tilde{\mu}_{t-1}(x)|^2 = |\Phi(x)(\Sigma_{t-1}^{-1/2})\varepsilon|^2 = (\Phi(x)^\top\Sigma_{t-1}^{-1}\Phi)\|\varepsilon\|_2^2$. Thus, we are able to relate the expression to $\tilde{\sigma}(x)_{t-1} = \sqrt{\Phi(x)^\top\rho^2\Sigma_{t-1}^{-1}\Phi(x)}$.

As the norm of a multivariate vector of dimension $m$ is distributed with $\chi_m^2$ distribution to get the tail bound we use a appropriate tail bound. Namely, we use a result from [27] that $P(\|\varepsilon\|_2^2 \ge m+2x) \le \exp(-x)$, which yields that

$$P(\|\varepsilon\|_2^2 \le m + 2\log(1/\delta)) \ge 1 - \delta. \tag{75}$$

Consequently, $|\tilde{f}_t(x) - \tilde{\mu}_{t-1}(x)|^2 \le \frac{(m+2\log(1/\delta))}{\rho^2}\tilde{\sigma}_{t-1}(x)^2 = \tilde{\beta}_t\tilde{\sigma}_t(x)^2$ and taking the square root yields the result. $\square$

**Lemma 12** (Gaussian Anti-concentration, [9]). *For a Gaussian variable $X$ with mean $\mu$ and variance $\sigma^2$, for any $\beta > 0$,*

$$P\left(\frac{X - \mu}{\sigma} > \beta\right) \ge \frac{e^{-\beta^2}}{4\sqrt{\pi}\beta}. \tag{76}$$

**Lemma 13.** *Let $k$ be a kernel s.t. $k(x, x) = 1$ and the observation to construct the confidence bounds be corrupted by noise with variance $\rho^2$, the the posterior variance is lower bounded by $\sqrt{\frac{\rho}{\rho+t}}$.*

*Proof.* The posterior standard deviation is $\sigma_t(x) = \sqrt{k(x,x) - k_t(x)^\top(\mathbf{K}_t)^{-1}k_t(x)}$. The variance at one point $x$ can decrease the most if all observations up till time $t$ were evaluated at $x$ only. Hence the matrix $\mathbf{K}_t = \mathbf{S} + \mathbf{I}\rho^2$ and $k_t(x) = 1$ are vector of full ones, where $\mathbf{S}$ is a matrix full of ones. Using basic algebra, we obtain that

$$\sigma_t(x)^2 \geq 1 - 1^\top(\mathbf{S} + \mathbf{I}\rho^2)1 = \frac{\rho}{\rho + t},$$

where the last equality follows from Propositions for $\rho$-matrix in [33]. $\qquad\square$

### D.7 Thompson sampling - II

**Lemma 14.** *Let Assumption 3 and $\Phi(\cdot)^\top\Phi(\cdot)$ $\epsilon$-uniformly approximate the kernel $k, \epsilon < 1, \delta_y \in (0, e^{-1})$, and denoting the approximated quantities with tilde then,*

$$P[\tilde{f}_t(x) \geq g(x)|\mathfrak{F}_{t-1}] \geq (1 - \delta_y)p\exp\left(-2\sqrt{\frac{\rho + t}{\rho B_t}}\xi(\epsilon, t) - \frac{\rho + t}{\rho B_t}\xi(\epsilon, t)^2\right) \qquad (77)$$

*where $p = \frac{1}{4\sqrt{\pi}}$, where $1 - \delta_y$ comes from the Theorem 5.*

*Proof.*

$$P(\tilde{f}_t(x) > g(x)|\mathfrak{F}_{t-1}) = P\left(\frac{\tilde{f}_t(x) - \tilde{\mu}_{t-1}(x)}{B_t^{1/2}\tilde{\sigma}_{t-1}(x)} > \frac{g(x) - \tilde{\mu}_{t-1}(x)}{B_t^{1/2}\tilde{\sigma}_{t-1}(x)}|\mathfrak{F}_{t-1}\right)$$

We notice that this scaling by $B_t^{1/2} = \max(\beta_t^{1/2}, \tilde{\beta}_t^{1/2})$ combined with Lemma 11 allows us to apply Lemma 12, with the $\vartheta = \frac{g(x) - \tilde{\mu}_{t-1}(x)}{B_t^{1/2}\tilde{\sigma}_{t-1}(x)}$, as

$$P(\tilde{f}_t(x) > g(x)|\mathfrak{F}_{t-1}) \geq \frac{e^{-\vartheta^2}}{4\sqrt{\pi}\vartheta} \geq \frac{e^{-\vartheta^2}}{4\sqrt{\pi}}$$

where for the second inequality we assumed $\vartheta > 1$.

Further, Analyzing the expression,

$$\vartheta = |\frac{g(x) - \tilde{\mu}_{t-1}(x)|}{B_t^{1/2}\tilde{\sigma}_t(x)} \qquad (78)$$

$$\leq \frac{|g(x) - \mu_{t-1}(x)| + |\mu_{t-1}(x) - \tilde{\mu}_{t-1}(x)|}{B_t^{1/2}\tilde{\sigma}_t(x)} \qquad (79)$$

$$\leq \frac{B_t^{1/2}\sigma_{t-1}(x)}{B_t^{1/2}\tilde{\sigma}_{t-1}(x)} + \frac{\epsilon\frac{t^2}{\rho^2}(B + \sqrt{2\log(1/\delta_y)})}{B_t^{1/2}\tilde{\sigma}_{t-1}(x)} \qquad (80)$$

$$\leq \frac{\sigma_{t-1}(x)}{\tilde{\sigma}_{t-1}(x)} + \epsilon\sqrt{\frac{\rho + t}{t}}\frac{t^2(B + \sqrt{2\log(1/\delta_y)})}{\rho^2 B_t^{1/2}} \qquad (81)$$

where we used Theorems 8 and 5 for the second to last step, and Lemma 13 in the last step.

Analyzing the expression, we apply Proposition 1, and that $B_t^{1/2} \geq 1$ to get the final expression,

$$\vartheta \leq 1 + \sqrt{\frac{\rho + t}{\rho}}\sqrt{\epsilon}\left(\frac{2t^2}{\rho} + \frac{\sqrt{\epsilon}t^2(B + \sqrt{2\log(1/\delta_y)})}{\rho^2 B_t^{1/2}}\right) \leq 1 + \sqrt{\frac{\rho + t}{\rho B_t}}\xi(\epsilon, t) \qquad (82)$$

where we have use that $\sqrt{\epsilon} < 1$. $\qquad\square$

**Lemma 15.** *Let the $\Phi(\cdot)^\top\Phi(\cdot)$ $\epsilon$-uniformly approximate the kernel $k$, $\delta_y \in (0, e^{-1})$, and denoting the approximated quantities with tilde then the probability of picking a unsaturated arm is,*

$$P(x_t \in D \setminus S_t|\mathfrak{F}_{t-1}) \geq (1 - \delta_y)p\exp\left(-2\sqrt{\frac{\rho + t}{\rho B_t}}\xi(\epsilon, t) - \frac{\rho + t}{\rho B_t}\xi(\epsilon, t)^2\right), \qquad (83)$$

*where $p = \frac{1}{4\sqrt{\pi}}$, where $1 - \delta_y$ comes from the Theorem 5.*

*Proof.* Note that first that we know the following three inequalities,

$$|g(x) - \mu_{t-1}| \overset{\text{Theorem 8}}{\leq} B_t^{1/2}\sigma_{t-1}(x)$$

$$|\tilde{f}_t(x) - \tilde{\mu}_{t-1}| \overset{\text{Lemma 11}}{\leq} B_t^{1/2}\tilde{\sigma}_{t-1}(x)$$

$$|\mu_{t-1}(x) - \tilde{\mu}(x)| \overset{\text{Theorem 5}}{\leq} \epsilon\frac{t^2}{\rho^2}(B + \sqrt{2\log(1/\delta_y)})$$

Consequently,

$$
\begin{aligned}
|\tilde{f}_t(x) - g(x)| &\leq 2B_t^{1/2}\tilde{\sigma}_{t-1}(x) + B_t^{1/2}\frac{2t^2\sqrt{\epsilon}}{\rho} + \epsilon\frac{t^{3/2}}{\rho^2}(B + \sqrt{2\log(1/\delta_y)}) \\
&= 2B_t^{1/2}\tilde{\sigma}_{t-1}(x) + \xi(\epsilon, t). \tag{84}
\end{aligned}
$$

The point $x_t$ is chosen as a maximizer of $\tilde{f}_t$. Hence, if $x_t \in D \setminus S_t$, then the values of $\tilde{f}_t$ over $S_t$ must be larger namely over $x^*$,

$$\mathrm{P} \overset{\text{def}}{=} \mathrm{P}(x_t \in D \setminus S_t | \mathfrak{F}_{t-1}) \geq \mathrm{P}(\hat{f}_t(x^*) \geq \hat{f}_t(x) | x \in S_t, \mathfrak{F}_{t-1}).$$

We use first the (84) and then the fact that $x \in S_t$ to obtain the final result,

$$\mathrm{P} \geq \mathrm{P}(\hat{f}_t(x^*) \geq g(x) - 2B_t^{1/2}\tilde{\sigma}_{t-1}(x) - \xi(\epsilon, t) | x \in S_t, \mathfrak{F}_{t-1}) \geq \mathrm{P}(\hat{f}_t(x^*) \geq g(x^*) | \mathfrak{F}_{t-1}).$$

Applying the Lemma 14 finishes the result. $\qquad\square$

**Remark 2.** *Note, we can control the quantities in the above Lemma by lowering $\epsilon$, as $\epsilon \to 0$, $(1 - \delta_y)p\exp(-2\sqrt{\frac{\rho+t}{\rho}}\xi(\epsilon, t) - \frac{\rho+t}{\rho}\xi(\epsilon, t)^2) \to \frac{1-\delta_y}{4\sqrt{\pi e}}$, as in the analysis of non-approximated Thompson sampling [9].*

**Lemma 16.** *Let the $\Phi(\cdot)^\top\Phi(\cdot)$ $\epsilon$-uniformly approximate the kernel $k$, $\delta_y \in (0, e^{-1})$, and denoting the approximated quantities with tilde then the expected instantaneous regret is*

$$\mathbb{E}[r_t | \mathfrak{F}_{t-1}] \leq B_t^{1/2}(4p(\xi(\epsilon, t)) + 2)\mathbb{E}[\tilde{\sigma}_{t-1}(x_t) | \mathfrak{F}_{t-1}] + 3\xi(\epsilon, t), \tag{85}$$

*where $r_t = g(x^*) - g(x_t)$, and*

$$p(\xi(\epsilon, t)) \overset{\text{def}}{=} (1 - \delta_y)p\exp\left(-2\sqrt{\frac{\rho+t}{\rho B_t}}\xi(\epsilon, t) - \frac{\rho+t}{\rho B_t}\xi(\epsilon, t)^2\right), \tag{86}$$

*with probability $1 - \delta_y$*

*Proof.* Let us first define the following point,

$$\bar{x}_t = \arg\min_{x \in D\setminus S_t} \tilde{\sigma}_t(x). \tag{87}$$

Then, we split the analysis to two points

First we obtain a useful lower bound

$$
\begin{aligned}
\mathbb{E}[\tilde{\sigma}_{t-1}(x_t) | \mathfrak{F}_{t-1}] &\geq \mathbb{E}[\tilde{\sigma}_{t-1}(x_t) | x_t \in D \setminus S_t, \mathfrak{F}_{t-1}]\mathrm{P}(x_t \in D \setminus S_t) \tag{88} \\
&\quad + \mathbb{E}[\tilde{\sigma}_{t-1}(x_t) | x_t S_t, \mathfrak{F}_{t-1}]\mathrm{P}(x_t \in S_t) \\
&\geq \tilde{\sigma}_{t-1}(\bar{x}_t)p(\xi(\epsilon, t)). \tag{89}
\end{aligned}
$$

Secondly, let us consider the following,

$$
\begin{aligned}
r_t &= g(x^*) - g(x_t) = g(x^*) - g(\bar{x}_t) + g(\bar{x}_t) - g(x_t) \\
&\overset{(84)}{\leq} g(x^*) - g(\bar{x}_t) + \tilde{f}_t(\bar{x}_t) + 2B_t^{1/2}\tilde{\sigma}_{t-1}(\bar{x}_t) - \tilde{f}_t(x_t) + 2B_t^{1/2}\tilde{\sigma}_{t-1}(x_t) + 2\xi(\epsilon, t)
\end{aligned}
$$

$$\overset{\text{(87), } x_t \text{ is maximizer}}{\leq} \quad 4B_t^{1/2}\tilde{\sigma}_{t-1}(\bar{x}_t) + 2B_t^{1/2}\tilde{\sigma}_{t-1}(x_t) + 3\xi(\epsilon, t)$$

$$\overset{\text{(89)}}{\leq} \quad 4B_t^{1/2}p(\xi(\epsilon,t))\mathbb{E}[\tilde{\sigma}_{t-1}(x_t)|\mathfrak{F}_{t-1}] + 2B_t^{1/2}\tilde{\sigma}_{t-1}(x_t) + 3\xi(\epsilon,t)$$

$$\mathbb{E}[r_t|\mathfrak{F}_{t-1}] \overset{\text{Tower prop.}}{\leq} \quad B_t^{1/2}(4p(\xi(\epsilon,t)) + 2)\mathbb{E}[\tilde{\sigma}_{t-1}(x_t)|\mathfrak{F}_{t-1}] + 3\xi(\epsilon,t).$$

$\square$

**Remark 3** (Martingale sequence). *Let the $\Phi(\cdot)^\top\Phi(\cdot)$ $\epsilon$-uniformly approximate the kernel $k$, $\delta_y \in (0, e^{-1})$, and denoting the approximated quantities with tilde then let us define the following,*

$$X_t = r_t - B_t^{1/2}(4p(\xi(\epsilon,t)) + 2)\tilde{\sigma}_{t-1}(x_t) + 3\xi(\epsilon,t) \tag{90}$$

$$Y_t = \sum_{s=1}^{t} X_s. \tag{91}$$

*due to Lemma 16, $Y_t$ is a martingale.*

**Lemma 17.** *Let $\xi(\epsilon,t)$ be as in Definition 9 and $p(\xi(\epsilon,t))$ as in (86), then $\xi(\epsilon,t)$ is increasing function of $t$, and $p(\xi(\epsilon,t))$ is decreasing function of $t$ bounded by $\frac{1-\delta_y}{4\sqrt{\pi}}$ from below.*

*Proof.* The first statement follow from definition of $\xi(\epsilon,t)$. The second statement follows from the first statement, namely, $\exp(-2\sqrt{\frac{\rho+t}{\rho}}\xi(\epsilon,t) - \frac{\rho+t}{\rho}\xi(\epsilon,t)^2) \to 0$ as $t \to \infty$. $\square$

*Proof of Theorem 6.* Due to Remark 3, we know that the sum in (91) is a martingale. Consequently, we can apply Azuma-Hoeffding inequality. To apply it we need that $|Y_t - Y_{t-1}|$ is bounded. In our case, we it is bounded by $X_t$. Thus,

$$\text{P} \overset{\text{def}}{=} \text{P}\left(Y_T \leq \sqrt{2\log(1/\delta_1)\sum_{t=1}^{T}X_t^2}\right) \geq 1 - \delta_1.$$

We note that the fact that the problem is not solved to the optimality introduces a $\sqrt{T}$ factor to the regret which will be absorbed by the asymptotic notation.

We use the following notational shorthand $Q_T^{1/2} = B_T^{1/2}(\pi(1-\delta_y)+2)$ and $R_T = \sum_{t=1}^{T} r_t$. Consequently, substituting the results inside the above equation, and using $|r_t| \leq 2B$ due to boundedness assumption and Lemma 17,

$$1 - \delta_1 \leq$$

$$\text{P}\left(R_T \leq \sum_{t=1}^{T}Q_t^{1/2}\tilde{\sigma}_{t-1}(x_t) + 3\xi(\epsilon,t) + \sqrt{2\log\left(\frac{1}{\delta_1}\right)\sum_{t=1}^{T}\left(2B + Q_t^{1/2} + 3\xi(\epsilon,t)\right)^2}\right) \leq$$

$$\text{P}\left(R_T \leq 3\xi(\epsilon,T)T + Q_T^{1/2}\sum_{t=1}^{T}\tilde{\sigma}_{t-1}(x_t) + \sqrt{2T\log\left(\frac{1}{\delta_1}\right)\left(2B + Q_T^{1/2} + 3\xi(\epsilon,T)\right)^2}\right) \leq$$

$$\text{P}\left(R_T \leq 3\xi(\epsilon,T)T + Q_T^{1/2}\sqrt{T\gamma_T(\tilde{k})} + \sqrt{2T\log\left(\frac{1}{\delta_1}\right)\left(2B + Q_T^{1/2} + 3\xi(\epsilon,T)\right)^2}\right) \leq$$

$$\text{P}\left(R_T \leq \mathcal{O}\left(\sqrt{\epsilon}T^3 + B_T^{1/2}\sqrt{T\gamma_T(\tilde{k})}\left(1 + \log\left(\frac{1}{\delta_1}\right)\right)\right)\right) \overset{\text{(33)}}{\leq}$$

$$\text{P}\left(R_T \leq \mathcal{O}\left(\sqrt{\epsilon}T^3 + B_T^{1/2}\sqrt{Tm\log T}\right)\right)$$

$$B_T^{1/2} = \mathcal{O}\left(\max(\sqrt{m}, \sqrt{\gamma_T(k)})\right)$$

Using Corollary 3, we can provide a bound on maximum information gain of additive squared exponential kernel. Thus, the final bound becomes,

$$\mathrm{P}\left(R_T \leq \mathcal{O}\left(\sqrt{\epsilon}T^3\log(1/\delta) + \log(1/\delta)^{3/2}\max\left(m\log T\sqrt{T}, G\sqrt{Tm(\log T)^{\bar{d}+2}}\right)\right)\right). \quad (92)$$

We take union on the probability that Azuma inequality holds, each round of sampling the Lemma 11 ($\delta$ inside $B_T$), Theorem 5 ($\delta_y$) and Theorem 8 holds ($\delta$ inside $B_T$). Thus, the final result with $\delta_1 = \delta_y = \delta$ holds with $1 - (3+T)\delta$). Rearranging with the new delta $\tilde{\delta} = \frac{\delta}{T}$ finishes the proof.

$\square$

## Footnotes

[1]The original version that appeared stated a wrong lemma on $\chi^2$ concentration which has been corrected without changing the results qualitatively.