[Reviews · NeurIPS 2018]

Reviewer 1



In this paper, the authors introduce an improvement to the standard random fourier features (RFF) scheme for kernel approximation. Rather than rely on Monte Carlo estimates of the integral involved in Bochner's theorem, the authors derive features from a Gauss-Hermite quadrature estimate of the integral. The result is a kernel estimation scheme that convergences uniformly over the domain quickly. The authors demonstrate theoretically that the improved uniform convergence results in theoretical guarantees on the convergence of both the posterior moments of the Gaussian process and the cumulative regret of Bayesian optimization. Overall, I quite like the paper. It tackles an issue that really does come up in practice: I have often found random Fourier features to produce unsatisfactory sample paths from the posterior Gaussian process. A method that solves this issue both in practice and provides theoretical convergence guarantees is quite nice. In my opinion, this is good work. I do have a few questions and comments for the authors that I detail below. Right now, my biggest complaint is that the paper largely ignores the existence of approximate GP methods that aren't based on finite basis expansions. Inducing point methods (e.g. Titsias et al., 2009) can lead to accelerated sampling procedures. When performing stochastic variational inference (Hensman et al., 2013, Wilson et al., 2016), the time complexity required to perform sampling can be as low as O(tm^2) depending on the kernel approximation used. Pleiss et al., 2018 introduces a procedure that achieves O(t) time approximate sampling and applies it to the additive BayesOpt setting. As far as I am aware, theory does not exist demonstrating a cumulative regret bound for Bayesian optimization using sparse GP methods, but theory does exist demonstrating convergence of the posterior moments. I assume QFF is compatible with other acquisition functions that can be computed using sample paths from the posterior Gaussian process (e.g., max value entropy search (Wang et al., 2017), predictive entropy search (Hernández-Lobato et al., 2014), knowledge gradient (Scott et al., 2009), ...). Have you run experiments with these? The max value entropy search paper in particular discusses another (albeit fairly coarse approximate) approach to sampling directly from the maximum value distribution, p(y*|D). Finally, can the authors comment a bit on their focus on the high dimensional Bayesian optimization setting? In my mind, the primary contribution of the paper is the provably good kernel approximation that solves a number of issues with the RFF approach. Is this story largely due to a decreased applicability of QFF to high dimensional sampling? Is the regret bound in Theorem 3 substantively different as a result of using QFF? It appears to have the same flavor as has been demonstrated before for additive Bayesian optimization, where cumulative regret depends linearly on the number of additive components and exponentially on the size of the largest additive component. If QFF is limited to usage with additive structure GPs, it's worth noting that this setting also appears in deep kernel learning (DKL, Wilson et al., 2016) where it is standard practice to assume additive structure at the end of the neural network. QFF could therefore be used as a drop-in replacement for kernel approximation schemes in DKL.

Reviewer 2



This paper presents Quadrature Fourier Features as an alternative for random features to improve the scalability of Bayesian optimization under the additive model assumption. The additive model assumption has been extensively studied previously as correctly mentioned in the paper. In this case, the authors use a variation of the Thompson sampling acquisition function, where the sample is selected from the approximation of the GP model using QFF, which are much more efficient. However, as pointed out by the authors, although QFF scale better than the GPs for a high number of data points, it scale poorly with the dimension of problem. Given that the paper is about high dimensional BO, this is a serious concern. It is true that, for some problems, the effective dimension is lower, however, in the paper, all the experiments rely on a effective dimension of 1 or 2. This is completely unrealistic in practice. Given the limitations of QFF with the dimensionality of the problem, it is necessary to evaluate how far can be applied in practice. Minor comments: -Figs 2 and 3 are tiny and difficult to read even in the PDF. The legend is too invasive. Given that the colors are the same for multiple plots, they could use a single legend outside all plot or describe the colors in the caption. ---- The author's response about the effective dimensionality is compelling. It would be nice to have an experiment that pushes the limit of the method, like having multiple cliques with effective dimensionality of 4-6 each of them. Nevertheless, the paper is a nice step towards high-D BO.

Reviewer 3



The authors introduced a very nice approaximation approach with Quadrature Fourier features and a novel algorithm for high-dimensional BO with additive GPs. What's even nicer is that the algorithm comes with a regret bound that considered both the kernel approaximation and two different BO acquisition schemes: GP-UCB and Thompson sampling. The authors also made a step further by addressing the difficulty of optimizing the acquisition functions, which is one of the hardest problem in BO. I am very impressed overall. For improvements, I think the authors can - add more real-world functions for experiments; - add more intuitive explanations to Theorem 2 and 3, especially on the assumptions; - state the regret bound of GP-UCB in the main paper bt squeezing the big O notation: log T can be combined in Eq. (9), no? Minor problems: - It is good to label the dimensionality of variables, e.g. on line 186, maybe write \Phi \in R^{m\times n} or something like that - What is going on with the axes of Fig. 2? More minor, but important problems: The authors did not check the references' formats and whether they are correct. For example: - [8] was published at ICML 2012. Please check other references that were marked arxiv; they might have been published in proceedings of some conferences. It is important to cite correctly and not use the raw Google scholar info. - 'gaussian' in all the references should be 'Gaussian', and you probably need to write something '{G}aussian' in the bib file; Google scholar usually won't give you the right bib entry. The same for 'bayesian', which should be 'Bayesian'. Conference names also need to be capitalized.. - [9] does not have a conference/jounal/tech report. Please check again. - Please unify the conference names: if you use COLT, then use ICML, NIPS too. Otherwise, use the full conference names for everyone of them. I think these problems are easily fixable, so it might be a good idea to make sure those easily fixable details are good too before submitting to any conferences or arxiv. Typos: - Line 198, sets of points - Line 308, real data -> a real-world function